# Clinical and genomic features of Lynch syndrome differ by tumor site and disease spectrum

Shisen Li [1,6], Ningning Luo [2,6], Gaoxin Jin [3], Tiantian Han [2], Xiangyu Yin [4,5], Didi Guo [2], Xing Zhang[2,7] ✉ & Zhaobang Tan [1,7] ✉

Lynch Syndrome (LS) carriers occasionally develop central nervous system (CNS) malignancies or tumors in organs not traditionally linked to the syndrome. These tumors are poorly characterized in the literature, and there is no sufficient consensus on guidelines and management recommendations for these tumors. Here we study LS from the tumor perspective and profile 238 pan-cancer specimens from 228 genetically confirmed LS carriers. Tumors are stratified into CNS LS-related, non-CNS LS-related, and non-CNS LS-unrelated groups according to anatomic site and established LS tumor spectrum. Comparative analyses against TCGA reveal significant alterations in LS incidence within endometrial and hepatic cancers. Across the three groups, we reveal marked heterogeneity in germline pathogenic-variant distribution, age at diagnosis, somatic mutation landscapes, tumor mutational burden, and microsatellite-instability status. This site- and spectrum-based stratification of a large, pan-cancer LS cohort underscores the heterogeneity of the LS and provides a data-driven foundation for refining future disease management strategies.

Lynch syndrome (LS, MIM: 120435) is one of the most prevalent autosomal dominant hereditary cancer susceptibility syndromes in humans and is caused by alterations in one of four mismatch repair (MMR) genes, namely, *MLH1, MSH2, MSH6* and *PMS2*, or a structurally intact MSH2 epigenetic inactivation caused by a large deletion in the non-MMR gene epithelial cellular adhesion molecule (*EPCAM*)[1,2]. The MMR system plays a crucial role in maintaining genomic stability and preventing cancer within a biological system by rectifying errors arising during DNA replication and recombination[2,3]. Inactivation of the second allele typically occurs due to small pathogenic variants and gene loss, resulting in an impairment of the MMR system. DNA mismatch repair deficiency (dMMR) inactivates the ability to repair DNA

mismatches, inducing accelerated carcinogenesis[4]. LS increases the risk of developing several cancers throughout life, including cancers of the colon, rectum, endometrium, stomach, small bowel, biliary tract, pancreas, renal pelvis and/or ureter, bladder, kidney, ovary, brain, or prostate, which are known as LS-associated cancers because of their significantly greater frequency compared to that of the average population[5-7].

Tumors that occur in LS can develop at any age but often arise in young people, often have a family history, and have the possibility of developing multiple synchronous and metachronous tumors[8,9]. dMMR in LS often results in high-level microsatellite instability (MSI-H) and the absence of MMR protein(s) in tumor tissue, which provides

[1]Department of Digestive Surgery, Xijing Hospital of Digestive Diseases, Fourth Military Medical University, Xi'an, Shanxi, China. [2]The Medical Department, Jiangsu Simcere Diagnostics Co., Ltd., Nanjing Simcere Medical Laboratory Science Co., Ltd., The State Key Laboratory of Neurology and Oncology Drug Development, Nanjing, Jiangsu, China. [3]Department of Gastrointestinal Surgery, The People's Hospital of Jimo.Qingdao, Qingdao, Shandong, China. [4]Department of Biological Sciences, Xi'an Jiaotong-Liverpool University, Suzhou, China. [5]Institute of Infection, Veterinary & Ecological Sciences, University of Liverpool, Liverpool, UK. [6]These authors contributed equally: Shisen Li, Ningning Luo. [7]These authors jointly supervised this work: Xing Zhang, Zhaobang Tan. ✉ e-mail: xing.zhang@simceredx.com; ausgstudy@163.com

valuable shortcuts for identifying patients with LS[1]. The United States Food and Drug Administration (FDA) has now approved the use of immune checkpoint inhibitors for solid tumors exhibiting MSI-H or dMMR, making them a viable treatment option for LS-associated tumors[10,11]. Next-generation sequencing (NGS) of tumors provides an alternative method for LS screening[12,13]. Tumor screening and clinical history are both helpful in identifying patients who may have LS, but the identification of heterozygous pathogenic or likely pathogenic (P/LP) variants in one of the MMR genes (or *EPCAM*) is the definitive diagnosis of LS[2]. Importantly, with the wide application of NGS technology and multigene testing, the spectrum of LS-associated tumors has also continued to expand, which is beneficial for monitoring and risk management of patients with LS[14–16]. The tumor screening, identification, surveillance and therapy of patients with LS, as well as surgical management of colorectal and gynecological cancers, have been recognized by consensus guidelines[17–24].

Although LS quadruples the risk of brain tumors occurring, brain tumors are relatively rare and represent the uncharacterized tumor type in LS, and there is no sufficient consensus on guidelines and management recommendations[17–27]. Among central nervous system (CNS) tumors, glioma is the most commonly associated with LS, and medulloblastoma and neuroblastoma have also been reported; *MSH2* pathogenic variants are more common than other MMR mutations in CNS tumors[28]. In addition, patients with LS can also develop LS-unrelated cancers, such as breast cancer and sarcomas, because the data are insufficient to demonstrate that the risk of developing these cancers is increased in individuals with LS[5,14,29–32]. Age, MSI level, genomic alterations in tumors and other characteristics may differ among LS patients with different tumor types; thus, additional studies are needed to address the features of LS patients with noncanonical tumors[14]. Although some non-CNS LS-unrelated tumors and CNS LS-related tumors have been reported to develop in a few patients with LS and have been published as case reports and small retrospective cohorts, systematic studies with large sample sizes lack the description of the comprehensive characteristics of the population, especially comparisons with non-CNS LS-related tumors.

Here we study LS from the tumor perspective and investigate the similarities and differences in clinical and genomic features across multiple cancer types in CNS LS-related and non-CNS LS-unrelated tumor groups compared with those in the non-CNS LS-related tumor group, which will improve LS diagnostics, prevention, management and therapy.

## Results

### Patient enrollment

**The clinical characteristics of the enrolled patients with LS.** A total of 238 samples from 228 pan-cancer patients with LS were enrolled and divided into 3 groups according to the tumor location and tumor spectrum of the LS: the CNS LS-related tumor group (n = 68), the Non-CNS LS-related tumor group (n = 117), and the Non-CNS LS-unrelated tumor group (n = 53). The clinical characteristics of the three groups are shown in Table 1. There were significant differences in age among the three groups, with the age of the CNS LS-related tumor group being lower than that of the other two groups (p < 0.001). Germline variant gene (p < 0.001) and MSI status (p < 0.001) were also significantly different, but there was no significant difference in gender distribution among the three groups.

Figure 1A–D shows the gender and age distributions of the whole cohort and the three groups. In the overall population, there were fewer females than males (female: male = 103:135), and overall age showed a normal distribution, with the peak of both sexes appearing at 51–60 years of age. The age distributions of the three groups were different, especially in the CNS LS-related tumor group. The proportion of patients under 20 years old in the CNS LS-related tumor group was significantly greater than that in the other two groups (18/68, 1/117,

**Table 1 | Characteristics of the tumors of enrolled patients with Lynch syndrome in the three groups**

| | CNS LS-related tumor (N = 68) | Non-CNS LS-related tumor (N = 117) | Non-CNS LS-unrelated tumor (N = 53) | p-value |
|---|---|---|---|---|
| Age | | | | <0.001 |
| Mean (SD) | 38.0 (21.6) | 52.8 (13.2) | 55.5 (10.2) | |
| Median [Min, Max] | 36.5 [0, 77.0] | 54.0 [19.0, 79.0] | 57.0 [27.0, 76.0] | |
| Gender | | | | 0.992 |
| Female | 29 (42.6%) | 51 (43.6%) | 23 (43.4%) | |
| Male | 39 (57.4%) | 66 (56.4%) | 30 (56.6%) | |
| Germline variant gene | | | | <0.001 |
| *MLH1* | 10 (14.7%) | 37 (31.6%) | 5 (9.4%) | |
| *MSH2* | 29 (42.6%) | 31 (26.5%) | 9 (17.0%) | |
| *MSH6* | 17 (25.0%) | 27 (23.1%) | 16 (30.2%) | |
| *PMS2* | 12 (17.6%) | 22 (18.8%) | 23 (43.4%) | |
| MSI status | | | | <0.001 |
| MSI-H | 17 (25.0%) | 82 (70.1%) | 3 (5.7%) | |
| MSI-L | 10 (14.7%) | 2 (1.7%) | 1 (1.9%) | |
| MSS | 41 (60.3%) | 28 (23.9%) | 44 (83.0%) | |
| Unknown | 0 (0%) | 5 (4.3%) | 5 (9.4%) | |

Differences in mean age among the three groups were assessed using one-way analysis of variance (ANOVA) (data are presented as mean ± SD). The association between group and gender, germline variant gene, and MSI status was analyzed using Pearson's chi-squared test
*CNS* central nervous system, *LS* lynch syndrome, *SD* standard deviation, *MSI* microsatellite instability, *MSI-H* MSI-high, *MSI-L* MSI-low, *MSS* microsatellite stability.

0/53), and the peak value appeared in the 31–40 years old range. The peak age of the two Non-CNS groups both appeared at 51-60 years, but the age began to increase significantly from 30 years in the Non-CNS LS-related group, while from 40 years in the Non-CNS LS-unrelated group, which revealed that the Non-CNS LS-unrelated group may have a later age of onset than the Non-CNS LS-related group.

**Incidence of LS in different cancer types.** In this study, LS-associated germline P/LP variations were detected in a total of 19 cancer types. The 3 most common cancers detected were all non-CNS LS-related tumors, namely, EC (5.68%), urothelial cancer (3.59%), and CRC (1.96%). The three cancers with the lowest detected proportions were lung cancer (0.18%), gastric cancer (0.15%) and breast cancer (0.13%). The proportion of patients with EC in our cohort was significantly greater than that in the Cancer Genome Atlas (TCGA) cohort (5.68% vs. 1.66%, p = 0.0047), while the proportion of patients with liver cancer was significantly lower than that in the TCGA cohort (0.27% vs. 1.33%, p = 0.011)[33]. The incidence ratio of each cancer type in the Simceredx cohort and TCGA cohort is shown in Fig. 1E and the Supplementary Data 1.

**Germline mutational feature analysis of the MMR gene**
**The proportion of different MMR genes in different groups.** The number of germline P/LP variant genes in the 238 LS samples is shown in Fig. 2A, including *MLH1* (n = 52), *MSH2* (n = 69), *MSH6* (n = 60), and *PMS2* (n = 57), which were classified as novel (n = 47, 19.75%) or previously identified (n = 191, 80.25%) mutations according to whether they were reported or included in the database. Novel mutations indicate that these mutations were neither reported in the literature nor included in the database. The proportions of germline P/LP variant genes in different groups are shown in Fig. 2B. The proportions of *MSH2* in the CNS LS-related tumor group, *MLH1* in the Non-CNS LS-related tumor group and *PMS2* in the Non-CNS LS-unrelated tumor

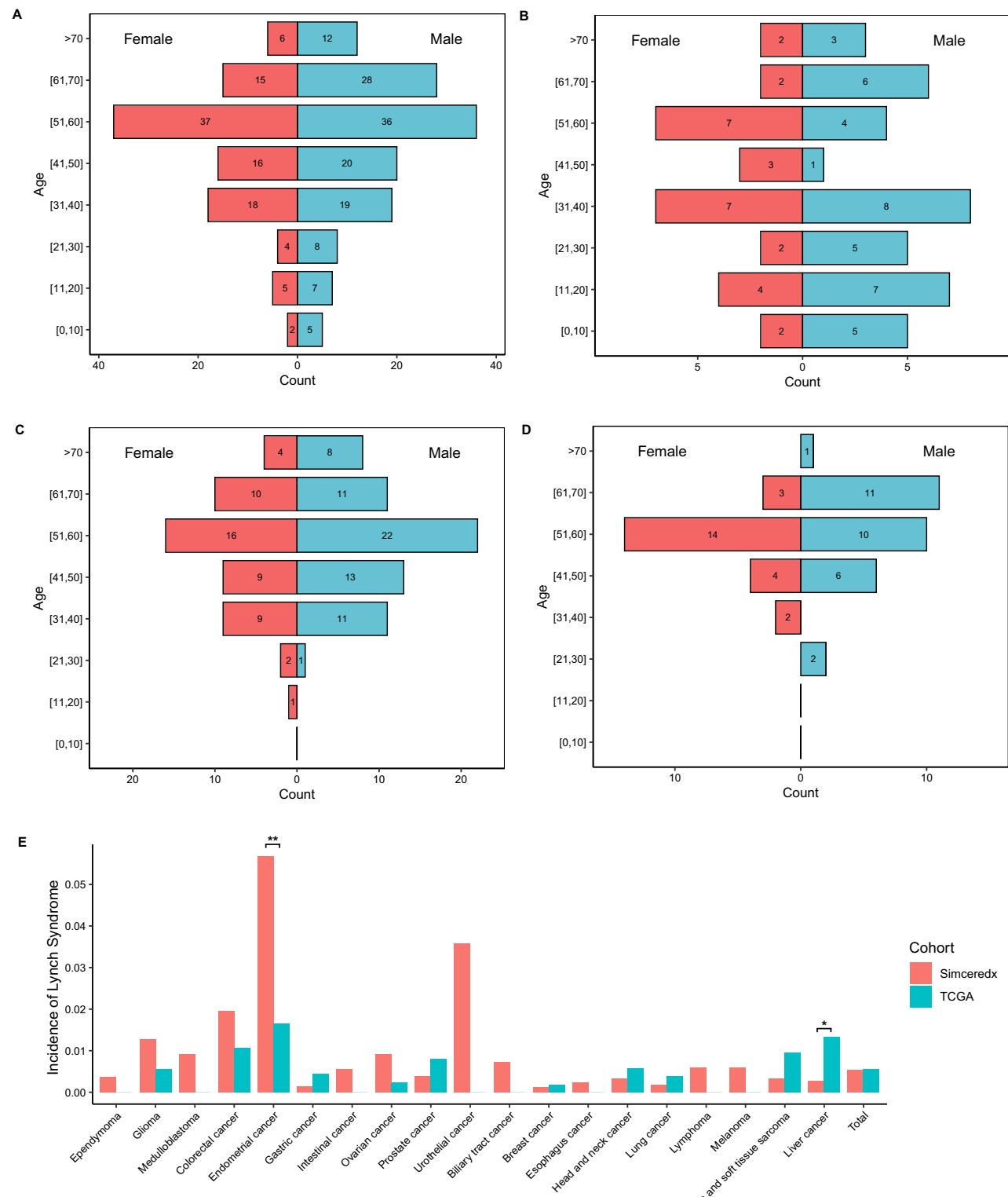

**Fig. 1 | The population pyramid of patients with Lynch syndrome (LS) and the distribution of different types of cancer. A**–**D** Gender and age distributions of 238 LS samples (**A**), Central nervous system (CNS) LS-related tumors (**B**), Non-CNS LS-related tumors (**C**), and Non-CNS LS-unrelated tumors (**D**). **E** Bar graph maps depicting the percentage of patients with LS in 19 different cancer types obtained from The Cancer Genome Atlas (TCGA) dataset versus the Simceredx dataset across 19 cancer types, and the overall difference. The number of patients with

lynch syndrome and overall, in the Simceredx cohort and the TCGA cohort are 477/83170 and 88/15812, respectively. Pearson's chi-squared test was used to evaluate the statistical difference of the incidence of lynch syndrome between the Simceredx cohort and the TCGA cohort (*$p < 0.05$; **$p < 0.01$; ***$p < 0.001$). Significant difference was observed in endometrial cancer (5.68% in Simceredx vs. 1.66% in TCGA, $p = 0.0047$, $\chi^2 = 7.9845$), and liver cancer (0.27% vs. 1.33%, $p = 0.011$, $\chi^2 = 6.4781$). Source data are provided as a Source Data file.

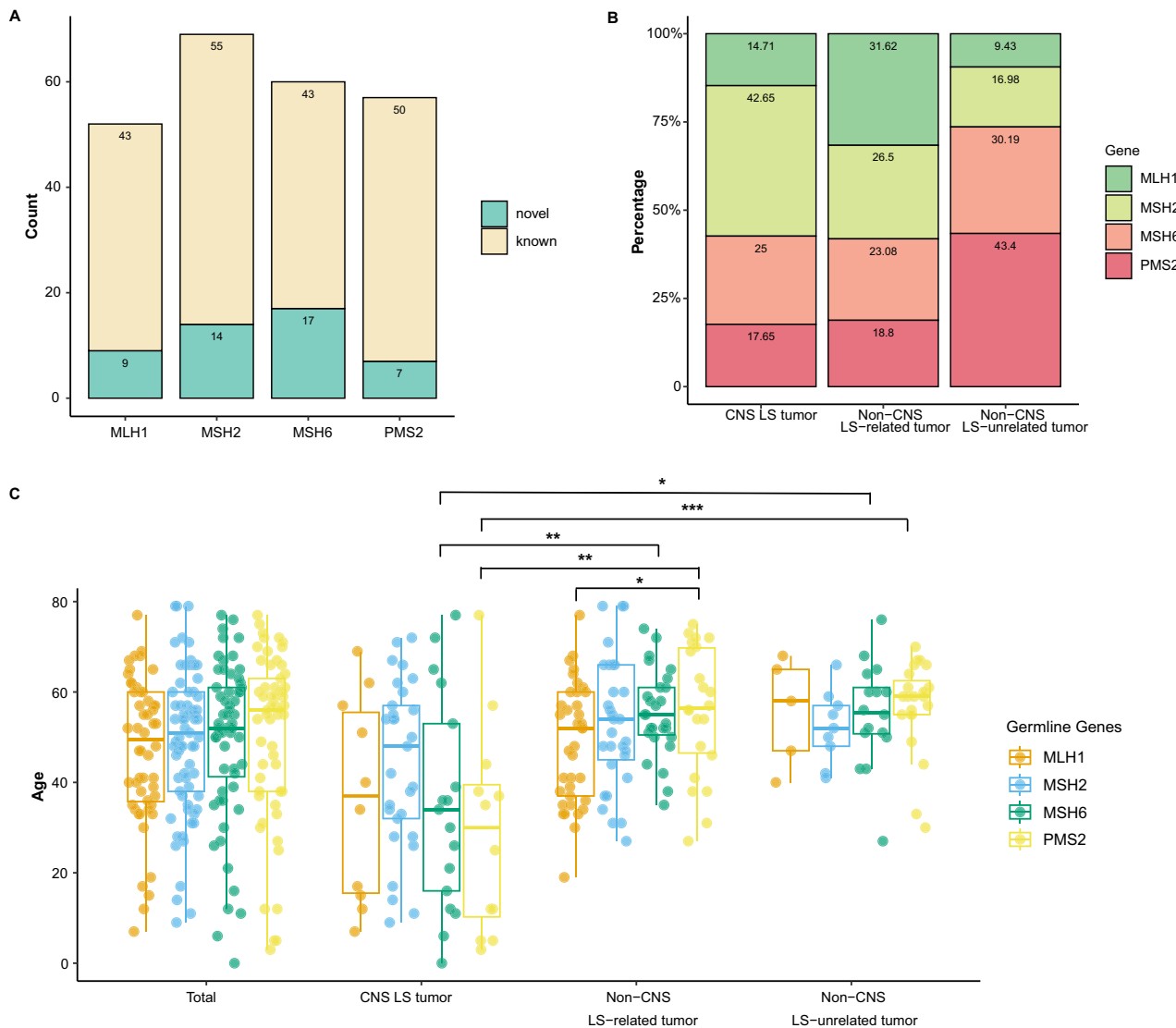

**Fig. 2 | Germline mutational features of mismatch repair (MMR) genes among different cohorts. A** Bar graph showing the number of germline pathogenic or likely pathogenic (P/LP) variants in MMR genes across 238 samples. **B** The proportions of different germline P/LP variant genes within the three cohorts. **C** Box plot showing the age distribution of different MMR carriers across distinct groups. A total of 238 samples, CNS LS tumor (*n* = 68), Non-CNS LS-related tumor (*n* = 117), and Non-CNS LS-unrelated tumor (*n* = 53) were included in this analysis, with each sample representing a single biological replicate. The box plot displays the median (center line), interquartile range (IQR) between 25th and 75th percentiles (bounds of the box), and whiskers extending to 1.5 times the IQR, and the points beyond are outliers, with "minima" and "maxima" as the absolute lowest and highest values. Colors denote the specific germline gene mutation, including *MLH1* (orange), *MSH2* (blue), *MSH6* (green), and *PMS2* (yellow). The *p*-values of comparisons were evaluated by a two-sided Mann–Whitney test, with multiple comparisons adjusted by the false discovery rate (FDR) approach. The asterisks denote significance levels: *$p < 0.05$, **$p < 0.01$, and ***$p < 0.001$. Source data are provided as a Source Data file.

group were the highest. There was no significant difference between the four genes in the CNS LS-related tumor group and the Non-CNS LS-related tumor group. The detection ratio of *PMS2* in the Non-CNS LS-unrelated tumor group was significantly greater than that in the CNS LS-related tumor group (43.4% vs. 17.65%, *p* = 0.036) and Non-CNS LS-related tumor group (43.4% vs. 18.8%, *p* = 0.021). The percentage of *MLH1* carriers was significantly greater in the Non-CNS LS-related group than in the Non-CNS LS-unrelated tumor group (31.62% vs. 9.43%, *p* = 0.021). The detection ratio of *MSH2* was significantly greater in the CNS LS-related tumor group than in the Non-CNS LS-unrelated tumor group (42.65% vs. 16.98%, *p* = 0.043). There was no significant difference in the proportion of *MSH6* among the three groups.

Lollipop plots of four gene variant sites on the peptide sequence in total and in the different groups are shown in Supplementary Fig. 1. *MLH1* is dominated by loss-of-function (LOF) mutations, including

frameshift, nonsense, and splicing variants. The mutations had no hot spot distribution area and were evenly distributed throughout the gene. The *MSH2*, *MSH6*, and *PMS2* genes were also similar.

**Age distribution of different germline pathogenic variant genes in different groups.** Box plots were drawn to explore the correlation between different germline P/LP variant genes and age in different groups (Fig. 2C). The results revealed that the age of patients harboring *PMS2* germline P/LP variants in the Non-CNS LS-related tumor group was significantly greater than that of patients harboring *MLH1* germline P/LP variants (*p* = 0.0499), but beyond that, there was no significant difference in age between the 4 P/LP variant genes in the same group. Comparison of the same gene between different groups revealed that there was no significant difference in the age distribution of patients harboring *MLH1* and *MSH2* germline P/LP variants among

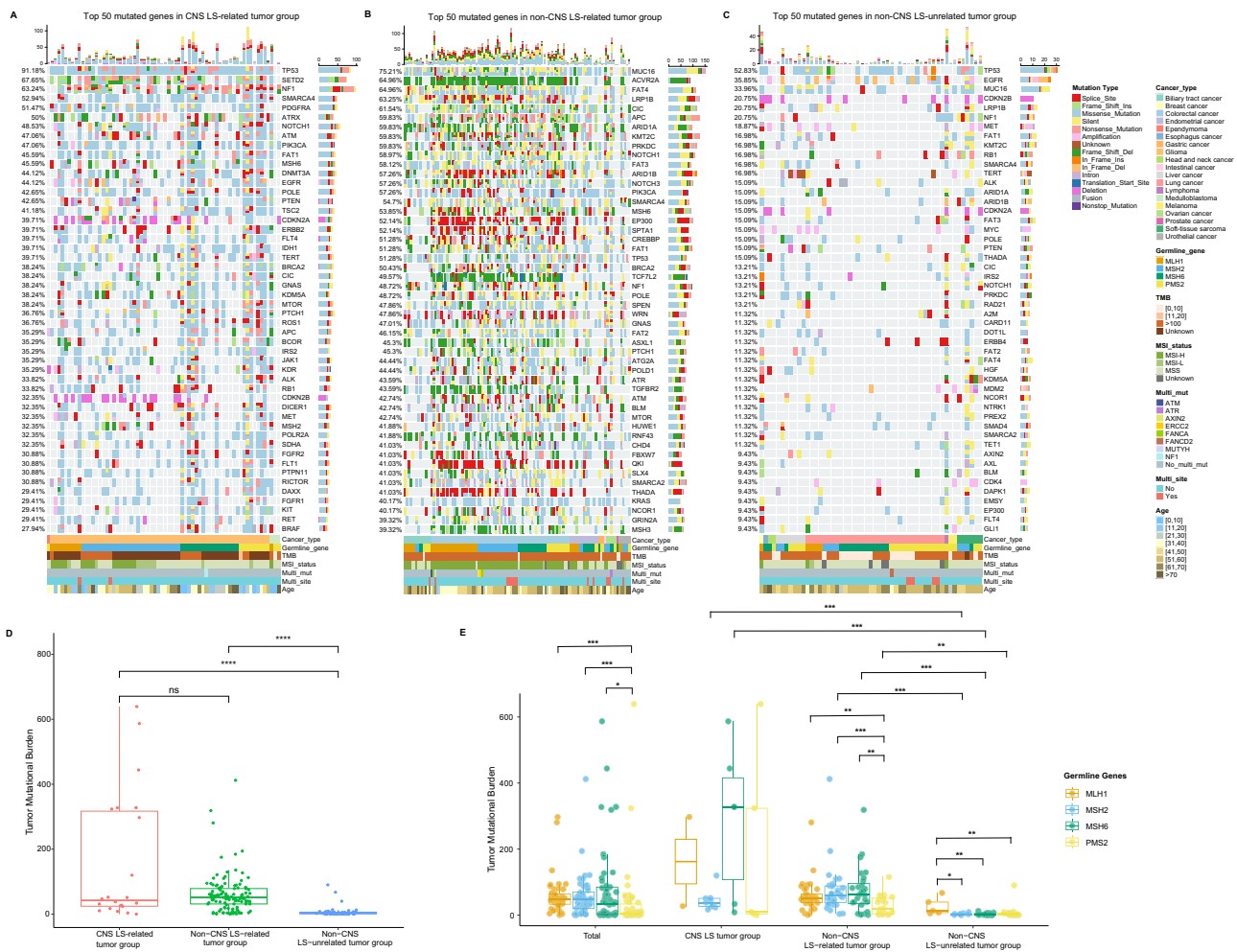

**Fig. 3 | The landscape plot and tumor mutation burden (TMB) analysis of patients with LS. A−C** Somatic mutation heatmaps of CNS LS-related tumors (**A**), non-CNS LS-related tumors (**B**), and non-CNS LS-unrelated tumors (**C**). Distinct colors correspond to specific mutational functions and clinical information, as indicated. Each column represents an individual patient, with the top 50 genes and mutation frequencies listed on the y-axis. **D, E** Box plot showing the comparison of TMB across the three groups (**D**) and the TMB of different germline mutant genes in different groups (**E**). A total of 191 patients have detected TMB values, with n = 22 in CNS LS tumor, n = 117 in Non-CNS LS-related tumor, and n = 52 in Non-CNS LS-unrelated tumor. The box plot displays the median (center line), interquartile range (IQR) between 25th and 75th percentiles (bounds of the box), and whiskers extending to 1.5 times the IQR, and the points beyond are outliers, with "minima" and "maxima" as the absolute lowest and highest values. Colors denote the specific germline gene mutation, including *MLH1* (orange), *MSH2* (blue), *MSH6* (green), and *PMS2* (yellow). The p-values of comparisons were evaluated by a two-sided Mann−Whitney test, with multiple comparisons adjusted by the FDR approach. (ns, p > 0.05; *p < 0.05; **p < 0.01; ***p < 0.001). Source data are provided as a Source Data file.

the three groups, while the age of patients harboring *MSH6* (p = 0.005; p = 0.016) and *PMS2* (p = 0.0019; p < 0.001) germline P/LP variants in the CNS LS-related tumor group was significantly lower than that in the other two groups.

In brief, we found that tumor samples in the three groups exhibited heterogeneity in the distribution of germline pathogenic genes (Fig. 2A, B), and there were significant differences in age among the three groups, with the age of the CNS LS-related tumor group being lower than that of the other two groups, especially in patients harboring *MSH6* and *PMS2* germline P/LP variants (Table 1 and Fig. 2C).

**Somatic mutational landscape and feature analysis**
**The somatic mutation heatmaps of the three groups.** To elucidate the genomic characteristics of LS, we conducted further analysis on the landscape of somatic mutations, TMB, and MSI status across the three groups. To explore the differences at the somatic gene mutation level among the three groups, we generated heatmaps of the top 50 genes in the three groups (Fig. 3A–C). The genomic mutation profiles

of the three groups were different, with the largest number of mutations in the Non-CNS LS-related tumor group and the smallest number of mutations in the Non-CNS LS-unrelated tumor group. The mutation frequency top 5 genes in the CNS LS-related tumor group were *TP53* (AF 91.18%), *SETD2* (AF 67.65%), *NF1* (AF 63.24%), *SMARCA4* (AF 52.94%) and *PDGFRA* (AF 51.47%); in the Non-CNS LS-related tumor group were *MUC16* (AF 75.21%), *ACVR2A* (AF 64.96%), *FAT4* (AF 64.96%), *LRP1B* (AF 63.25%) and *CIC* (AF 61.54%); and in the Non-CNS LS-unrelated tumor group were *TP53* (AF 52.83%), *EGFR* (AF 35.85%), *MUC16* (AF 33.96%), *CDKN2B* (AF 20.75%) and *LPR1B* (AF 20.75%).

**TMB in different groups and in different germline variant genes.** The box plots show the differences in TMB among the three groups (Fig. 3D). The median TMB was 42.27 (0–639.01), 51.47 (0.71–412.06) and 3.55 (0–90.07) in the CNS LS-related tumor group, Non-CNS LS-related tumor group and Non-CNS LS-unrelated tumor group, respectively. The TMB in the Non-CNS LS-unrelated tumor group was significantly lower than that in both the Non-CNS LS-related

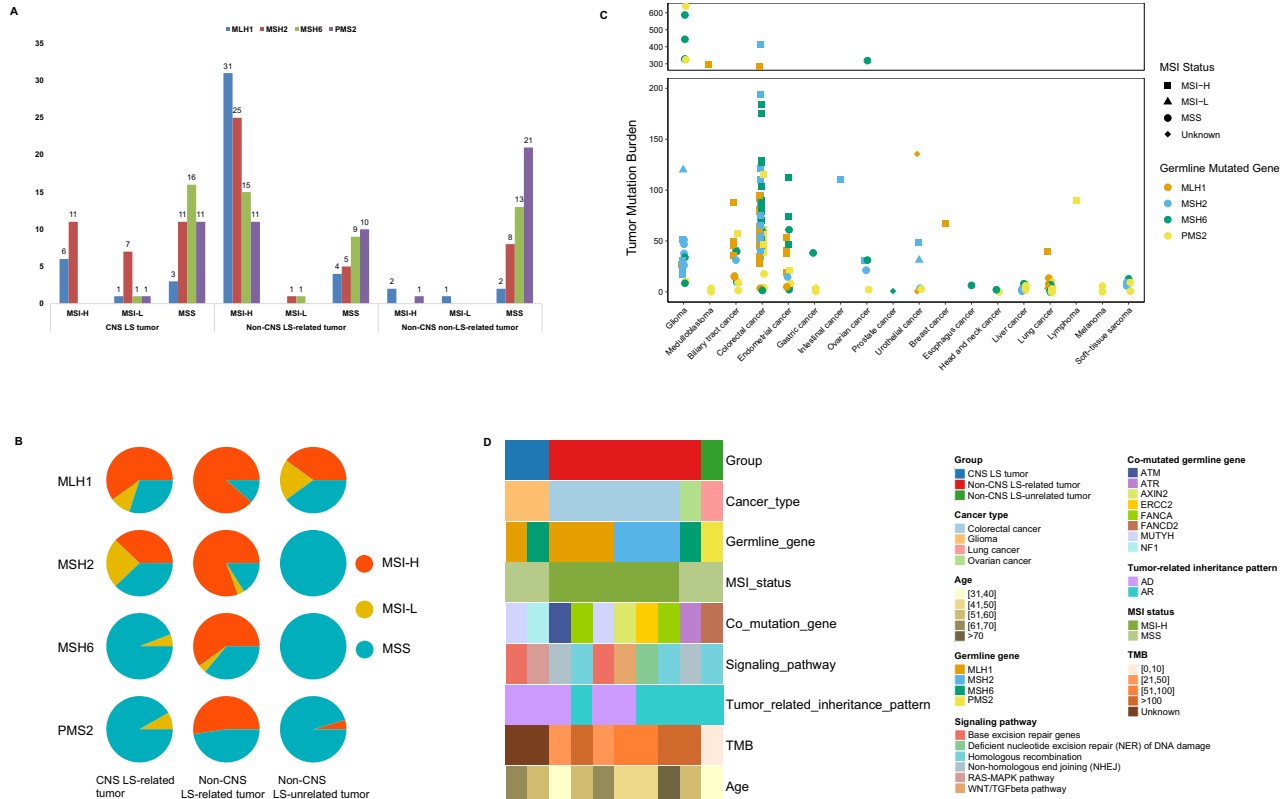

**Fig. 4 | Distribution patterns of microsatellite instability (MSI) status of different genes in different groups and characteristics of patients with LS harboring germline co-occurring susceptibility mutations. A** The numbers of different MSI status in different germline genes among the three groups are depicted, with distinct colors representing various MMR genes. The sample number is indicated on the y-axis. **B** Pie chart indicating the distribution of MSS tumors (blue), MSI-H tumors (dark orange) and MSI-L tumors (yellow) with respect to different germline genes among the three cohorts. **C** The TMB, MSI and germline MMR gene signatures of different cancer types for each sample were analyzed. The tumor types are indicated on the x-axis. TMB is indicated on the y-axis. The shape indicates the MSI status. The colors indicate the germline genes. **D** The characteristics of patients with LS harboring germline co-occurring susceptibility mutations. The different colors correspond to different characteristics, as indicated. Each column represents an individual patient. Source data are provided as a Source Data file.

tumor group (*p* < 0.001) and the CNS LS-related tumor group (*p* < 0.001).

To better explore the differences in TMB between the germline P/LP variant genes in different groups, we constructed box plots of the TMB of different germline variant genes in different groups (Fig. 3E). Intragroup comparison revealed that the TMB of patients harboring the other three genes was greater than that of patients harboring *PMS2* in the total population and in the non-CNS LS-related tumor group (*p* < 0.05). In the Non-CNS LS-unrelated tumor group, the TMB of patients harboring the *MLH1* gene was greater than that of patients harboring other genes (*p* < 0.05). The results showed that the TMB of samples with *MSH2* and *MSH6* germline variants in the CNS LS-related tumor and Non-CNS LS-related tumor groups was greater than that in the Non-CNS LS-unrelated tumor group (*p* < 0.001), and the TMB of samples with *PMS2* germline variants in the Non-CNS LS-related tumor group was greater than that in the Non-CNS LS-unrelated tumor group (*p* < 0.005).

These results suggest that the TMB features in the CNS LS-related tumor group are more similar to those in the Non-CNS LS-related tumor group.

**MSI status in different groups and in different germline variant genes.** The corresponding MSI status of samples with different germline pathogenic variant genes in different groups is shown in Fig. 4A, B. Compared with the other two groups, the Non-CNS LS-related tumor group had a greater proportion of MSI-H tumors,

especially in the samples with *MLH1* and *MSH2* germline P/LP variants. The proportion of MSI-H tumors in samples with *MLH1* variants was the highest, followed by that in samples with *MSH2* variants. In the Non-CNS LS-unrelated tumor group, MSH-H only existed in some samples with *MLH1* variants and in 1 patient with a *PMS2* variant. In the CNS LS-related tumor group, no samples with MSI-H status were found among the samples harboring *MSH6* and *PMS2*.

Figure 4C shows the TMB, MSI and germline variant genes in different cancer types for each sample with TMB information. In the non-CNS LS-unrelated tumor group, the TMB was generally low. In the CNS LS-related tumor group, 6 glioma samples (PLS041, PLS046, PLS124, PLS143, PLS185, PLS228) had a TMB greater than 300 but were MSS, and we found each patient harboring mutations in the DNA polymerase genes, *POLE* or *POLD1*, which are associated with a hypermutated phenotype. The count and statistical results of the MSI status corresponding to different genes in each group are shown in Supplementary Data 2.

**The exceptional cases in LS**
**Co-occurring germline susceptibility mutations in LS.** We identified 10 patients with LS harboring germline co-occurring susceptibility mutations in genes other than MMR genes. These co-occurring mutated genes were primarily involved in the DNA damage repair pathway, including *ATM, ATR, AXIN2, ERCC2, FANCA, FANCD2, MUTYH, NF1* (Fig. 4D). Notably, seven cases occurred in the non-CNS LS-related tumor group, including 6 patients with CRC exhibiting characteristics

of MSI-H and TMB-H. In contrast, the remaining three patients in the CNS LS-related tumor group and the non-CNS LS-unrelated tumor group had MSS. Intriguingly, two patients in the CNS LS-related tumor group were over 50 years old, suggesting that co-occurring autosomal dominant tumor-related genes, such as *NF1* variants, may drive tumorigenesis in these patients (Fig. 4D).

Our analysis of tumor germline susceptibility genes provides valuable insights into the underlying drivers of disease development in patients with LS, aiding in personalized management and avoiding diagnostic and treatment biases.

**Multiple primary cancers in LS**. Among the patients in our cohort, 9 (3.9%) were diagnosed with multiple primary tumors, with 19 samples. Table 2 shows the characteristics of LS patients with multiple primary cancers. Two-thirds (66.7%) of these patients had tumors in two different organs, while one-third (33.3%) had tumors in the same organ (e.g., the intestine or lung). Notably, 44.4% (4/9) of the patients with multiple primary tumors had different MSI status at different lesion sites, and all of these patients had tumors in two different organs. Patients with multiple primary tumors were older (average age: 55.2 years) than those with a single primary tumor. Additionally, females (*n* = 6) may have a greater probability of developing multiple primary tumors than males (*n* = 3). Understanding the characteristics of multiple primary tumors associated with LS is critical for accurate diagnosis, treatment planning, and management of the disease.

## Discussion

Our study represents a large-scale study of LS patients in China, including a total of 229 patients with LS in 19 cancer types identified from 44,040 pan-cancer samples. In this study, 238 final enrolled samples from 228 patients were divided into 3 groups according to tumor location and tumor spectrum of LS: the CNS LS-related tumor group (*n* = 68), the Non-CNS LS-related tumor group (*n* = 117), and the Non-CNS LS-unrelated tumor group (*n* = 53). We studied LS from the tumor perspective and analyzed clinical and genomic characteristics and the similarities and differences among the 3 groups. The exploration of age, incidence, germline alterations, and somatic mutation characteristics in different groups is conducive to improving LS identification, management and treatment.

The characteristics of the non-CNS LS-related tumor group in our study were similar to those reported in previous studies, with incidence rates of EC (5.68%) and CRC (1.96%)[34–36], a mean age of 52.8 years[9], a proportion of *MLH1* and *MSH2* genes with germline variants being the highest[5,34], a median TMB being high (51.47, from 0.71 to 412.06), and a ratio of MSI-H being 70.1%[8]. The median ages of the carriers of the *MLH1*, *MSH2*, *MSH6* and *PMS2* genes were 49.5, 51, 52, and 56 years, respectively, and the patients harboring *PMS2* germline P/LP variants were significantly older than the patients harboring *MLH1* germline P/LP variants (Fig. 2C), indicating that the trend of the age distribution in the Non-CNS LS-related tumor group was consistent with previous reports[5]. The TMB of patients with variants in the other three MMR genes was greater than that of patients with *PMS2* variants in the total population and in the non-CNS LS-related tumor group.

CNS tumors, as an extracolonic manifestation of LS, appear to be relatively infrequent, with the occurrence risk quadrupling, and the cumulative lifetime risk of brain tumors is significantly greater in individuals with *MSH2* mutations than in those with variants in other MMR genes[28,37,38]. In the CNS LS-related tumor group in our study, the *MSH2* gene (42.65%) accounted for the greatest proportion of the four MMR germline P/LP genes, which is consistent with previous reports and highlights the importance of screening for the *MSH2* gene in CNS tumors[25,28]. Both the mean age (38 years) and the age of patients harboring *MSH6* and *PMS2* germline P/LP variants in the CNS LS-related tumor group were significantly lower than those in the other two groups, while there was no significant difference in the age distribution

among the 4 MMR genes in the CNS LS-related tumor group. The *TP53* gene was the most frequently somatically mutated gene in the CNS LS-related tumor group, with an AF of 91.18%. There was no significant difference in the TMB distribution between the CNS LS-related tumor group and the Non-CNS LS-related tumor group, indicating that the TMB features were more similar between the two groups. However, in the CNS LS-related tumor group, MSI-H tumors were found only in samples with *MLH1* (6/10) and *MSH2* (11/29) germline P/LP variants. The CNS LS-related tumor group had a lower proportion of MSI-H tumors than the Non-CNS LS-related tumor group. This finding is consistent with previous reports indicating that MMRDness scores vary across different tissues, with brain tissue typically exhibiting lower scores compared to blood and gastrointestinal tissues[39]. Compared to non-CNS LS-related tumors, CNS-LS-related tumors are relatively rare and may manifest at an earlier age but have similar characteristics of MMR deficiency and have some slight genetic heterogeneity. Our research serves as a fundamental reference for the diagnosis and treatment application of CNS-LS-related tumors.

In the Non-CNS LS-unrelated tumor group in our study, the proportion of patients with *PMS2* (43.4%) was the highest, and this ratio was significantly greater than that in the other two groups; however, the proportion of patients with *MLH1* was significantly lower in the Non-CNS LS-unrelated group than in the Non-CNS LS-related group. It is widely believed that individuals carrying the P/LP *PMS2* variant have a much lower risk of developing cancer than other MMR carriers, although individual cancer risk may be influenced by various factors[40,41]. In the previous study[42], *PMS2* gene had the highest carrier rate among the four MMR genes in colorectal cancer cases. In our study, due to differences in the classification of cancer types and potential variations in ethnicity, *PMS2* gene did not have the highest proportion in the non-CNS LS-related tumor group, which includes colorectal cancer. LS patients may develop sporadic tumors not associated with MMR deficiency. These tumors may represent background cancer risk unrelated to LS. Our tumor-based analysis describes features of tumors arising in LS carriers but does not imply that every tumor is causally related to LS. *PMS2* variants have lower penetrance and higher prevalence in the general population. In the LS-unrelated group, the prevalence probably more resembles the prevalence of pathogenic MMR mutations as seen in the normal population, in which *PMS2* variants are most prevalent. The predominance of *PMS2* in the LS-unrelated group likely reflects background distribution, and most of these tumors might indeed be "LS-unrelated" in biology. This fact does not exclude that tumors in the LS-unrelated group are caused by MMR deficiency, but it is not unlikely that in most cases the tumors ack a second hit and are MMR-proficient. Future studies are needed to clarify which tumors in Lynch syndrome carriers are causally linked to mismatch-repair deficiency and which are not.

The mean age of the Non-CNS LS-unrelated tumor group was 55.5 years, and the proportion of patients with LS among all cancer types in this group was low, with a maximum of 0.6%. This group had the smallest number of mutations, and the TMB in this group was significantly lower than that in the other two groups. In addition, the TMB of samples with *MSH2* and *MSH6* germline variants in this group was lower than that in the other two groups, and the TMB of *PMS2* germline variant samples in this group was lower than that in the non-CNS LS-related tumor group. The TMB of patients harboring the *MLH1* gene was greater than that of patients harboring other genes in this group. MSSs accounted for 83% of this group, and MSH-H existed mostly in samples with *MLH1* variants. Non-CNS LS-unrelated tumors exhibit molecular features (late onset, low TMB/MSS) consistent with sporadic cancers not driven by MMR deficiency, which indicates that these tumors may be truly unrelated to LS.

People with LS should receive lifelong cancer screening starting in adulthood, and should be counseled in the proper way, irrespective of a cancer diagnosis or diagnosed tumor type. Surveillance of non-CNS

**Table 2 | Characteristics of LS patients with multiple primary cancers**

| Patient number | Gender | TMB | MSI | Group | Cancer type | Germline gene | hgvs_c | hgvs_p | Tumor location |
|---|---|---|---|---|---|---|---|---|---|
| PLS052 | Male | 58.16 | MSI-H | Non-CNS LS-related tumor | Colorectal cancer | MSH2 | c.1753dup | p.S585Ffs*13 | Colon |
| PLS052 | Male | 109.93 | MSI-H | Non-CNS LS-related tumor | Colorectal cancer | MSH2 | c.1753dup | p.S585Ffs*13 | Ileocecum |
| PLS076 | Female | 0.0 | MSS | Non-CNS LS-unrelated tumor | Lung cancer | PMS2 | c.1997_1998del | p.K666Rfs*4 | Right upper lung |
| PLS076 | Female | 0.0 | MSS | Non-CNS LS-unrelated tumor | Lung cancer | PMS2 | c.1997_1998del | p.K666Rfs*4 | Right middle lung |
| PLS076 | Female | 4.26 | MSS | Non-CNS LS-unrelated tumor | Lung cancer | PMS2 | c.1997_1998del | p.K666Rfs*4 | Right lower lung |
| PLS080 | Male | 4.26 | MSS | Non-CNS LS-unrelated tumor | Lung cancer | PMS2 | c.1 A > G | p.M1? | Lung |
| PLS080 | Male | 9.22 | MSS | Non-CNS LS-related tumor | Biliary tract cancer | PMS2 | c.1 A > G | p.M1? | Cholecyst |
| PLS087 | Female | 21.28 | MSS | Non-CNS LS-related tumor | Ovarian cancer | MSH2 | c.2038 C > T | p.R680* | Endometrium |
| PLS087 | Female | 31.21 | MSI-H | Non-CNS LS-related tumor | Ovarian cancer | MSH2 | c.2038 C > T | p.R680* | Ovary |
| PLS103 | Female | 63.12 | MSI-H | Non-CNS LS-related tumor | Colorectal cancer | MSH2 | c.2194_2209dup | p.R737Nfs*18 | Colon |
| PLS103 | Female | 48.23 | MSI-H | Non-CNS LS-related tumor | Urothelial cancer | MSH2 | c.2194_2209dup | p.R737Nfs*18 | Bladder |
| PLS157 | Female | 46.81 | MSI-H | Non-CNS LS-related tumor | Endometrial cancer | MSH6 | c.3788_3801+8del | . | Uterus |
| PLS157 | Female | 2.13 | MSS | Non-CNS LS-related tumor | Endometrial cancer | MSH6 | c.3788_3801+8del | . | Uterus |
| PLS163 | Female | 412.06 | MSI-H | Non-CNS LS-related tumor | Colorectal cancer | MSH2 | c.387_388del | p.Q130Vfs*2 | Colon |
| PLS163 | Female | 51.06 | MSS | CNS LS tumor | Glioma | MSH2 | c.387_388del | p.Q130Vfs*2 | Brain |
| PLS210 | Female | 75.18 | MSI-H | Non-CNS LS-related tumor | Colorectal cancer | MSH2 | c.858del | p.F286Lfs*6 | Colon |
| PLS210 | Female | 54.61 | MSI-H | Non-CNS LS-related tumor | Colorectal cancer | MSH2 | c.858del | p.F286Lfs*6 | Colon |
| PLS218 | Male | 34.75 | MSI-H | Non-CNS LS-related tumor | Colorectal cancer | MLH1 | c.928dupA | p.T310Nfs*4 | Colorectum |
| PLS218 | Male | N/A | MSS | CNS LS tumor | Glioma | MLH1 | c.928dup | p.T310Nfs*4 | Brain |

LS Lynch syndrome, TMB tumor mutation burden, MSI microsatellite instability, MSI-H MSI-high, MSS microsatellite stability, CNS central nervous system.

LS-related tumors, such as CRC, EC, and urothelial cancer, has been recommended for individuals with LS at an early age[16]. However, there is not enough evidence to support increased screening above average risk in Non-CNS LS-unrelated tumor screening recommendations. Our findings suggest that patients with non-CNS LS-unrelated tumors should not be grouped together with patients with non-CNS LS-related tumors when considering genetic counseling or clinical management, and the later onset of non-CNS LS-unrelated tumors may also justify specific guidelines for surveillance that are tailored to this tumor type. Revised international clinical guidelines for MMR carriers for different tumor groups should be based on a comprehensive assessment of their associated cancer risks.

LS is an autosomal dominant genetic disorder linked to a high risk of cancer and appears to be common in multiple ethnicities[1]. We compared the occurrence ratio of different cancer types in the Simceredx cohort to that in the TCGA cohort. The proportion of patients with EC in our cohort was significantly greater than that in the TCGA cohort, while the proportion of patients with liver cancer was significantly lower than that in the TCGA cohort, which may be related to ethnicity[33]. Significant differences in genomic mutation profiles, mutation types and genes with high frequencies of mutations among the three groups were also observed. This observation implies that LS exhibits significant heterogeneity, indicating that its diverse pathogenesis is driven by distinct genetic alterations. The observed differences in age at onset, MSI, and TMB can serve as pilot data for future risk-prediction models and guideline revisions. This study offers valuable reference points for future mechanistic investigations and clinical decision-making, providing a robust hypothesis-generating foundation for both basic research and prospective trials. To improve the management of CNS LS-related tumors and non-CNS LS-unrelated tumors, further comprehensive multiomics analyses are needed.

In the present study, we found that ~4.4% (10/229) of LS patients had co-occurring pathogenic variants in non-MMR hereditary cancer genes. These co-occurring mutations were predominantly associated with the DNA damage repair pathway. The identification of a second pathogenic mutation in the cancer susceptibility gene is an unexpected finding. Most hereditary cancer syndromes are Mendelian disorders characterized by complex phenotypes due to the influence of additional independently inherited genetic variations and/or environmental factors[43]. An increasing number of clinical phenotypic overlaps among distinct inherited cancer syndromes have been documented[29,44]. Among the patients included in our cohort, 9 were diagnosed with multiple primary tumors. Notably, a significant proportion of these patients (2/3) exhibited tumors in distinct organs. Individuals with LS exhibit an elevated lifetime risk of developing various malignancies and multiple primary cancers compared to the general population, and the risk of multiple secondary tumors in patients with LS is of concern[9,45]. For patients diagnosed with LS, multigene panel testing of multiple primary tumors is recommended to ascertain the underlying etiology of the neoplasms.

There are several limitations to our study. First, our study used DNA-based NGS tests, which have some limitations in the detection of some types of variation, such as large genome rearrangements (LGRs) and epigenetic changes in MMR genes, although these variants are rare but often associated with the etiology of LS[46,47]. Second, constitutional mismatch repair deficiency (CMMRD) has been identified in individuals with biallelic pathogenic germline mutations in MMR genes[48,49]. In our study, among the 20 patients under the age of 25, two patients had two germline variants in the same MMR gene (PLS081 *PMS2* P and LP; PLS227 *MLH1* LP and VUS, respectively), which strongly suggests a diagnosis of CMMRD. It has been reported that primary mismatch repair-deficient gliomas in children, adolescents, and young adults represent a distinct patient group with similar characteristics that can benefit from immune checkpoint inhibitor therapy[50]. Given the small number of only two cases with an extremely low population incidence

of CMMRD, we have not excluded them at this stage. In the future, the newly published method[39,51] can be considered to collect more confirmed cases of CNS CMMRD and LS according to the current differential diagnosis recommendations of CMMRD[52], and explore and distinguish their characteristics, which have important clinical value. However, our study was retrospective and did not include family verification, so the diagnosis of CMMRD of the two patients could not be conclusively confirmed.

In conclusion, our study investigates the clinical and genomic characteristics and the similarities and differences among CNS LS-related tumors, non-CNS LS-related tumors, and non-CNS LS-unrelated tumors, which hold paramount importance for patient management and precision-based therapeutic interventions.

## Methods

### Patient enrollment

Our study was approved by the Ethical Committee of Xijing Hospital of Digestive Diseases, Fourth Military Medical University (KY20252370-F-1) and the Institutional Review Board of Nanjing Simcere Medical Laboratory Science (NSML-IRB-202406-MS03). All adult participants or guardians provided written informed consent. Neither sex nor gender was considered in the study design, since the primary focus of this study was unrelated to sex or gender. Participants received no compensation.

A total of 229 patients diagnosed with LS harboring heterozygous germline P/LP variants in at least one of the MMR genes (*MLH1, MSH2, MSH6,* and *PMS2*) were enrolled in this study. The 239 tumor samples from these 229 LS patients included 19 samples from 9 patients with multiple primary cancers. One sample from a patient with cancer of unknown primary origin was excluded. Considering that samples from different sites in patients with multiple primaries might be assigned to different groups, 238 samples from 228 LS patients were ultimately enrolled and used for all subsequent analyses. Baseline characteristics information, including age, gender and cancer types of all patients, was collected. Among the 228 enrolled patients, 132 were male and 96 were female, with a median age of 52.5 years (range: 0–79).

### DNA extraction, library preparation and genomic sequencing

Frozen fresh tumor tissues or formalin-fixed paraffin-embedded tumor slides of 238 tumor samples and paired blood samples were collected and sequenced at Nanjing Simcere Diagnostics Laboratory (Simceredx, Nanjing, China), the College of American Pathologists (CAP, Certificate ID: 1856000) and Clinical Laboratory Improvement Amendments (CLIA, Certificate ID: 99D2289380) co-accredited laboratories, using hybridization capture-based targeted NGS of the 551-gene panel (*n* = 181), 131-gene panel (*n* = 46), or the 353-gene panel (Neuro-Onco360) (*n* = 11). The gene lists for each panel are provided in Supplementary Data 3.

A Concert® gDNA Tissue Extraction Kit was used to extract genomic DNA (gDNA) from tumor tissues, and gDNA from paired leukocytes was extracted using a magnetic gDNA kit (TIANGEN®). Then, the gDNA was quantified through a Qubit dsDNA HS Assay Kit with a Thermo Fisher Scientific Qubit Fluorometer, and its quality was evaluated using an Agilent 4200 TapeStation (Agilent). Two hundred nanograms of the above-qualified gDNA were enzymatically snipped into fragments of 200–300 bp. The ends of the gDNA fragments were then repaired using the KAPA Hyper DNA Library Preparation Kit (Roche Diagnostics). The VAHTSTM Universal DNA Library Prep Kit for Illumina® (Vazyme) was used to conduct A-tailing and to produce indexed paired-end adaptors for the SimcereDx Illumina platform. After eliminating unwanted contaminants using Agencourt AMPure XP beads (Beckman Coulter), a pre-library was generated by polymerase chain reaction (PCR).

The 150 bp paired-end sequencing of the final qualified DNA library was performed using the Illumina NovaSeq 6000 platform

according to the manufacturer's recommendations. The fastp software (V.2.20.0) can obtain various quality control indicators for each sample, trim the adapter, and eliminate low-quality bases[53]. The following step was the alignment of the paired-end read sequences with the UCSC hg19/GRCh37 reference genome using the Burrows–Wheeler Aligner (BWA-MEM v.0.7.17) algorithm.

## Bioinformatics analysis of sequencing data

Subsequently, we called and annotated the single-nucleotide variation (SNV) and insertion/deletion (Indel) mutations using VarDict (v.1.5.7) and InterVar tools, respectively[54,55]. We then screened public databases, such as variants in the 1000 Genomes Project (August 2015) and the Exome Aggregation Consortium Browser28, for common single-nucleotide polymorphisms. The fusion genes were accessed through Factera (v1.4.4)[56], while the copy number variations (CNVs) were obtained using CNVkit (dx1.1)[57].

The cutoff set for the minimal somatic variant allele frequency was 2%. The tumor mutation burden (TMB) was defined as the sum of nonsynonymous somatic mutations in the protein-coding region per megabase (muts/Mb). The MSI was determined based on the homopolymer repeat loci on the detected panel with sufficient coverage, and it was calculated as the percentage of unstable loci[58]. According to the thresholds, the MSI values were then transformed into MSI-high (MSI-H), MSI-low (MSI-L), and MS-stable (MSS) values.

Germline variants sequenced by paired blood samples were screened by VarDict (v.1.5.7) and then annotated to public databases, including gnomad (v3.1.2), CLINVAR (202308), dbNSFP (v42a), COSMIC (v98), and the Simceredx database. After filtration, the pathogenicity assessment for each mutation was performed according to the integrated outcomes of InterVar and CLINVAR databases. According to the American College of Medical Genetics and Genomics (ACMG) guidelines, germline mutations were stratified into five categories: P, LP, benign, likely benign, and variants of unknown significance (VUS), in which P and LP variants are typically selected for further analysis[59].

## Lynch syndrome analysis in The Cancer Genome Atlas (TCGA) cohort

We obtained data from the publication by Huang et al.[33], including information on cancer types, germline genes with P/LP variants, specific mutations and variant classifications and so on. From this dataset, we identified samples with germline variants in MMR genes, which are associated with LS. We then calculated the incidence of LS across different cancer types and further compared these incidences with those in our own cohort.

## Statistical analysis

R version 4.3.1 and Excel were used for all analyses. The top mutated somatic genes and germline variant genes were analyzed and visualized utilizing the R package maftools and ComplexHeatmap[60,61]. Other figures were generated using the R packages ggplot2 and Excel. The differences between the continuous variables were analyzed using the Wilcoxon test, while the chi-square test was chosen for the large sample analysis. The differences between different groups of the matrix pie charts were generated through Fisher's exact test. A $p$-value < 0.05 was considered to indicate statistical significance.

## Reporting summary

Further information on research design is available in the Nature Portfolio Reporting Summary linked to this article.

## Data availability

The genomic raw sequence data generated in this study have been deposited in the Genome Sequence Archive (Genomics, Proteomics & Bioinformatics 2025) in the National Genomics Data Center (Nucleic Acids Res 2025), China National Center for Bioinformation/Beijing Institute of Genomics, Chinese Academy of Sciences (GSA-Human: HRA013167), which are publicly accessible at https://ngdc.cncb.ac.cn/gsa-human. The raw sequence data are available under restricted access for the files containing potentially identifiable human genetic information. Chinese regulations on human genetic resources require that access be controlled; consequently, only bona fide investigators from academic or other non-profit institutions may obtain the dataset for non-commercial research that is consistent with the participants' consent, and any attempt to re-identify individuals is prohibited. Researchers who wish to access the data must log in to the GSA-Human portal, submit an online controlled-access application that includes a brief research plan, local ethical-approval documentation, and a signed institutional Data Transfer Agreement, and are encouraged to address any preliminary questions to the corresponding author, Xing Zhang. The GSA-Human Data Access Committee will evaluate complete requests and issue a decision within ten working days; for approved projects, the download credentials will remain active for 1 year. The germline mutation data of the TCGA database used in this study are available in Cancer Genome Atlas (https://www.cell.com/cell/fulltext/S0092-8674(18)30363-5). The remaining data are available in the Article, Supplementary Information, or Source Data file. Source data are provided with this paper.

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

## Acknowledgements

We thank all patients and their families who participated in this study. The authors thank Mr. Dongsheng Chen, Mr. Qin Zhang, Mrs. Xueyu Yang, Mr. Yuhao You and Mr. Wanglong Deng from Simceredx for the kind assistance.

## Author contributions

Z.T., X.Z., S.L., and N.L. conceived and designed this study; G.J., N.L., S.L., D.G., and T.H. collected samples and information; N.L. and X.Y. analyzed the data; S.L., N.L., T.H., and D.G. wrote the paper; X.Z. and Z.T. supervised the work. All authors read and approved the final manuscript.

## Competing interests

N.L., T.H., D.G., and X.Z. are current employees of Jiangsu Simcere Diagnostics Co., Ltd. The remaining authors declare no competing interests.
