## [Transparent Peer Review file · Nature Communications]

Clinical and genomic features of Lynch syndrome across distinct tumor spectrum groups implicates for practice

Corresponding Author: Professor Zhaobang Tan

Version 0:

Reviewer comments:

Reviewer #1

(Remarks to the Author)

Authors provide a solid review of Lynch Syndrome (LS) from a molecular and clinical perspective, summarizing its importance in cancer surveillance and treatment. They particularly wish to investigate CNS LS-related tumors, as well as non-LS-related tumors as well in a cohort of LS patients. This is an effective description of a cohort of LS patients in China, describing tumor types, age of onset, and tumor characteristics (mutations, TMB, MSI status).

I would remove on line 381 the sentence "Our findings suggest that PMS2 carriers also need to be concerned about the risk of non-CNS LS-unrelated tumors." All Lynch Syndrome patients should be concerned about the risk of LS-unrelated tumors. PMS2 is the most common LS gene to be mutated (PMID 27799157), and so it makes sense that they may end up having more non-CNS LS-unrelated tumors.

A major shortfall, that unfortunately will not be able to be corrected (they admit as much since their analysis was retrospective), is that they could not confirm that their young patients with CNS tumors were truly CMMRD. These are a distinct subset of LS patients with a different clinical course, particularly when it comes to CNS cancer risk vs LS; they are right to make note of it, but really would deserve their own subgroup for analysis.

[Copy edits: Line 234 = should be "carriers" not carries]

Reviewer #2

(Remarks to the Author)

This study describes a cohort of tumours diagnosed in a cohort of Lynch syndrome patents from China. Although the analysis of 238 tumours is relevant for the field, the manuscript can be certainly improved.

Please find below a list of comments/suggestions:

The overall contribution of the study is unclear, as well as its relevance to screening management within Lynch syndrome.

The clarity of the writing can be certainly improved. The terminology is very confusing in multiple instances:

incidence/prevalence, mutation frequency of the allelic gene, FASTQ document, healthy allele, super high TMB....

Abstract

The abstract lacks specificity in the presentation of numerical data, detailed methodology, and precise definitions, which limits the clarity and impact of the findings.

How is the prevalence of Lynch syndrome estimated from TCGA data?

Introduction

The introduction is too extensive and needs to be shortened. It does not help to understand the focus of the study and the specific objectives.

The topics in the introduction are not clearly defined, with concepts mixed in a way that makes it difficult to follow.

Please refer to relevant and updated bibliography (i.e. ref 32 and 33 cannot be used as current LS guidelines)

One relevant topic is the specific characteristics of MSI in CNS LS-associated tumors (PMID: 36240479). Please discuss the findings according to this previous knowledge throughout the manuscript.

Methodology

In general, the terminology is not accurate.

Patients and samples section lacks clarity. Please clarify the type of samples analyzed, how LS patients are identified, type of mutations (germline/somatic?), sample inclusion / exclusion, etc...

What is Simceredx database?

Please detail the genes included in the NGS analysis.

The bioinformatics analyses are hard to follow. It is not clear how they have used the germline and somatic data, and which strategy has been used for variant calling (tumor-normal paired, tumor-only ...), which genes/regions have been analyzed...

The terminology is not accurate. Lack of details on some tools that have been used (eg. MSI), filters and annotation.

Can TMB be calculated from panel data? Has it been previously validated?

Please specify the thresholds used for TMB and MSI categorization.

The TCGA cohort is not described.

It is not clear how they identify germline PV for identifying LS individuals (both in Sincerdex and TCGA cohorts)

Results

The results section and figures are redundant.

The titles of the results are not self-explanatory and do not provide clear information.

Section 1 ("Patient enrollment")

The reasoning behind the tumor classification into the three groups is not explained.

Why is the frequency of CNS-related LS tumors so high in the cohort?

The high proportion of CNS tumor patients under 20 suggests CMMRD. CMMRD can be discarded by measuring MSI in normal tissues by high sensitivity MSI analysis (i.e. PMID: 38531023)

It is unclear what is being compared to the TCGA, or how the LS patients in the cohort are being used in the analysis. TCGA data is not introduced.

Please revise axis title in Figure 1E.

Table 1 refers to tumors, not patients as stated in the title. Please clarify.

Section 2 ("Germline mutational feature analyses")

Very vague description of the germline mutational spectrum of the gene. Why silent variants have been considered?

Please clarify the meaning of "novel" and "non-novel" mutations.

This section needs to be summarized.

Section 3 ("Somatic mutational landscape and feature analyses")

Please clarify what the contribution of this analysis is.

Mutational data lacks clear interpretation of its clinical or biological significance. The comparison between groups is not well contextualized, and the writing could be more concise.

Please clarify how the 50 top genes were selected (the most mutated ones)? How have the significant differences between groups been assessed?

Please detail the specific mutations identified in the 50 top mutated genes. Are there recurrent variants?

Terminology of mutation type is not accurate, and no conclusion has been drawn from this data.

The non-CNS LS-unrelated tumors have low TMB. They are probably sporadic tumors. Is the identification of LS in this tumor samples well-assessed?

Regarding the 5 glioma samples with TMB>300 mut/Mb, have they somatic mutations in POLE/POLD1 genes?

Figure 4: it is not possible to see all the columns represented in the graph (i.e. PMS2 in the CMS LS-related tumors group, MSS)

Section 4 ("The exceptional cases in LS")

Hard to follow. Which germline mutations co-occur in the LS patients of the cohort?

Table 2: Does the table refer to multiple primary tumors? It is surprising that patient PLS076 harbors 3 lung tumors.

Discussion.

It needs to be extensively revised according to the previous comments.

Why MSI is present in non-LS associated tumors from MLH1 carriers?

Please clarify how obtained results impact on LS management strategies, as stated in the abstract.

Version 1:

Reviewer comments:

Reviewer #1

(Remarks to the Author)

Thank you very much for addressing reviewers' concerns with your revised manuscript. However, before publication, the following changes need to be made:

While I appreciate the changes made to reflect reviewers' comments in the abstract, the abstract background needs to be changed to read more cleanly, for example:

"Lynch Syndrome (LS) patients may sometimes develop cancers of the central nervous system (CNS) as well as in sites not traditionally associated with LS. These tumors are poorly characterized in the literature, and there are no sufficient consensus on guidelines and management recommendations for these tumors. Therefore, additional studies are needed to

elucidate the features of LS patients with noncanonical tumors.”

-Line 42 = what does finalized mean?

-Line 49 = should be “germline pathogenic variants”

-Line 420/421 = Remove “The incidence of primary tumors is relatively low in clinical practice” as this statement seems surplus to requirements; moreover, depending on one’s definition of cancer, 1 in 2 individuals ultimately develop some sort of cancer

Reviewer #3

(Remarks to the Author)

As a new reviewer on this manuscript, I was asked to primarily respond to the revisions made based on the comments raised by reviewer #2, but also to give general feedback on the study. Regarding the comments of reviewer #2, I acknowledge that the paper has been improved and many points raised have been properly addressed. There are, however, also some criticisms that remain unaddressed or incompletely addressed:

1. R#2: “The overall contribution of the study is unclear as well as its relevance to screening management within Lynch syndrome.” In my opinion, this aspect is only partly addressed, particularly regarding to screening guidelines. For defining strategies for screening it is crucial what the risk is for carriers of MMR mutations to develop a particular type of cancer. Such a study requires a perspective from the family or patient, not from the tumor. A characterization of tumors, as done in this study, provides information on relative differences of tumor types between the MMR genes and about the likelihood that the underlying germline mutations are indeed causative, for example by presenting with high TML or MSI. It is certainly true that the age at diagnosis of a particular tumor type may be helpful in developing criteria for screening, but whether a screening for a cancer type is needed requires data on the incidence of that cancer. Therefore, it is desirable that the authors are more precise on how their data contribute to improve LS surveillance.

2. R#2: “Please refer to relevant and updated bibliography (i.e. ref 32 and 33 cannot be used as current LS guidelines)”. This comment refers to refs 24 and 25 in the revision. I agree with the reviewer that these references are not optimal. Please check e.g. the ERN Genturix website for the latest published recommendations for LS.

3. R#2: “The high proportion of CNS tumor patients under 20 suggests CMMRD. CMMRD can be discarded by measuring MSI in normal tissues by high sensitivity MSI analysis (i.e. PMID: 38531023)”. This point is not properly addressed. There are two patients in their cohort with two germline mutations in a MMR gene. I do not agree with the authors that CMMRD is unlikely in these cases. In fact, with two (likely) pathogenic mutations in PMS2 it is very likely that this patient has CMMRD and also for the MLH1 biallelic carrier (P + VUS), it is highly suspected. These patients will certainly have high tumor mutational loads and, in case of a CNS tumor, will likely also have a somatic POLE or POLD1 on top of that (PMID 40368937). This should be visible in the data. The CMMRD tumor spectrum is different between CMMRD and LS, so every reason to take them out. The statement in the discussion that the two cases have “an extremely low probability of CMMRD” cannot be based on their data and is simply wrong.

4. I would not prefer the use of “non-novel” mutations. Previously identified/known would be better.

5. R#2: “Section 3 (“Somatic mutational landscape and feature analyses”) Please clarify what the contribution of this analysis is. Mutational data lacks clear interpretation of its clinical or biological significance.” The value of comparing mutation spectra of different tumor types with each other is not made clear. How can a comparison of heatmaps based on mixed groups of tumor types provide ‘deeper understanding of their potential associations with tumor development and progression.’? What is this deeper understanding? It is known that the spectrum of CNS tumors is different from non-CNS Lynch spectrum tumors, but there are also differences between endometrial and colorectal cancers. The tumors in the non-related group is even more diverse.

6. R#2: “Please detail the specific mutations identified in the 50 top mutated genes. Are there recurrent variants?” The authors state that it was not possible to identify recurrent mutations, but it is unclear why. Perhaps the question was misunderstood, but in general, the study lacks openness about core data on which they base their conclusions.

7. R#2: “The non-CNS LS-unrelated tumors have low TMB. They are probably sporadic tumors. Is the identification of LS in this tumor samples well-assessed?”. The reviewer raises an important point here, but the question may have caused confusion. Whereas the diagnosis of LS is beyond doubt is a reasonable assumption that the tumors are sporadic tumors, i.e. not caused/driven by MMR deficiency. LS families have higher risk of cancer, but obviously they also develop sporadic MMR-proficient cancers.

8. R#2: “Regarding the 5 glioma samples with TMB>300 mut/Mb, have they somatic mutations in POLE/POLD1 genes?” It is nice that these specific data are now provided in the manuscript, but this, again, illustrates the incomplete presentation of the data. It is impossible for readers to correlate individual findings, like type of germline findings, somatic mutations, TMB, MSI status etc. Raw data should be made available (through EGA submission) and detailed supplementary tables should be provided that allow readers to correlate independent findings to patient IDs. Several of the points raised by reviewer #2 originate from this underlying shortcoming.

In addition to the above comments, I believe that the findings presented in this study and the approach used in itself interesting. The study is descriptive but provides some new insights. Nevertheless, I do still have some major concerns:

- Data availability. I do not consider it acceptable that data are available upon request. Sequence data should be made available in EGA as is common practice in the current open science community. In addition detailed Tables are needed that couple all presented findings to patient IDs, to allow correlations between individual findings, which includes TMB, MSI status, germline mutations, driver mutations, etc to patient samples.

- The authors state in their discussion that ‘Our findings suggest that all LS patients should be concerned about the risk of LS-unrelated tumors, especially those carriers PMS2 gene.’ I believe that this statement is illustrative of a misconception in

this study. PMS2 variants are common in the normal population. What is the evidence that these tumors caused by LS? In LS, the distribution of MMR genes is biased towards MLH1 and MSH2 which are the genes with higher penetrance. In the LS-unrelated group, the prevalence probably more resembles the prevalence of pathogenic MMR mutations as seen in the normal population, in which PMS2 variants are most prevalent. This fact does not exclude that tumors in the LS-unrelated group are caused by MMR deficiency, but it is not unlikely that in the majority of cases the tumors lack a second hit and are MMR-proficient. This is also what is observed by the authors and, therefore, the conclusion should be that these tumors are truly LS-unrelated.

Version 2:

Reviewer comments:

Reviewer #3

(Remarks to the Author)

Thank you for addressing these points and clarifying the inconsistencies. Most of these issues are now more clearly formulated. As I indicated already, studying LS from the tumor perspective is interesting and important and the cohort is certainly informative, but it is essential that the difference between this approach and that of studies using the family/patient perspective is made clear to readers, and that the hypotheses and aims are adjusted to that.

There are a few small remarks remaining:

- Lines 229-302: "... harboring mutations in hypermutated POLE/POLD1 genes." Please rephrase, because the POLE/POLD1 genes themselves are not hypermutated

- Discussion (lines 409-411): "...that non-CNS LS-unrelated tumors should not be grouped together with non-CNS LS-related tumors for genetic counseling or clinical management". Here, it is again the wording that is not chosen well. When we talk about genetic counseling we talk about patients not tumors. The patients have LS and should be counseled in the proper way irrespective of a cancer diagnosis or diagnosed tumor type.

- Discussion (lines 445-456): "...which could not exclude the diagnosis of CMMRD.." still is a suboptimal way to phrase this. The fact that these patients had two LP/P variants strongly suggests a diagnosis of CMMRD. The phrasing 'does not exclude' suggests to readers that the authors have found something that argues against a CMMRD diagnosis, but without complete evidence.

RESPONSE TO REVIEWERS' COMMENTS

Reviewer #1 (Remarks to the Author):

Authors provide a solid review of Lynch Syndrome (LS) from a molecular and clinical perspective, summarizing its importance in cancer surveillance and treatment. They particularly wish to investigate CNS LS-related tumors, as well as non-LS-related tumors as well in a cohort of LS patients. This is an effective description of a cohort of LS patients in China, describing tumor types, age of onset, and tumor characteristics (mutations, TMB, MSI status).

I would remove on line 381 the sentence "Our findings suggest that PMS2 carriers need to be concerned about the risk of non-CNS LS-unrelated tumors." All Lynch Syndrome patients should be concerned about the risk of LS-unrelated tumors. PMS2 is the most common LS gene to be mutated (PMID 27799157), and so it makes sense that they may end up having more non-CNS LS-unrelated tumors.

Response: Thank you very much for your valuable suggestion. We have removed the sentence in line 381 and revised it as follows: " Our findings suggest that all LS patients should be concerned about the risk of LS-unrelated tumors, especially those carriers PMS2 gene." **Figures 2A and 2B** respectively illustrate the number of the four MMR genes and the proportion of each gene in the three groups. As can be seen from the figures, *PMS2* has the highest proportion in the non-CNS LS-unrelated tumor group. In the study referenced by [PMID 27799157], *PMS2* had the highest carrier rate among colorectal cancer cases. However, in our study, differences in cancer type classification and potential variations in ethnicity may have contributed to *PMS2* not having the highest proportion in the non-CNS LS-related tumor group, which includes colorectal cancer. We have also incorporated a discussion of this reference in the **Discussion** section of our paper:

In the previous study [PMID 27799157], *PMS2* gene had the highest carrier rate among the four MMR genes in colorectal cancer cases. In our study, due to differences in the classification of cancer types and potential variations in ethnicity, *PMS2* gene did not have the highest proportion in the non-CNS LS-related tumor group, which includes colorectal cancer. Our findings suggest that all LS patients should be concerned about the risk of LS-unrelated tumors, especially those carriers *PMS2* gene.

A major shortfall, that unfortunately will not be able to be corrected (they admit as much since their

analysis was retrospective), is that they could not confirm that their young patients with CNS tumors were truly CMMRD. These are a distinct subset of LS patients with a different clinical course, particularly when it comes to CNS cancer risk vs LS; they are right to make note of it, but really would deserve their own subgroup for analysis.

Response: Thank you very much for your reminder. We have reviewed the germline variants (including Pathogenic, Likely Pathogenic, and Variant of Uncertain Significance) of the mismatch repair (MMR) genes detected in all patients under the age of 25 (N = 20). Except for two patients who had two germline variants in the same MMR gene (*MLHI* LP/VUS and *PMS2* P/LP, respectively), the remaining 18 patients each had only a single P/LP mutation in a definite gene.

In our retrospective study, there was indeed no way to obtain additional samples to determine whether these two patients had LS or CMMRD. CMMRD is a distinct early-onset cancer predisposition syndrome (OMIM #276300). The estimated birth incidence is extremely low, with approximately one in a million if parents are not related. These two patients represent a small proportion of the overall cohort. In addition, in gliomas, primary mismatch repair-deficient gliomas in children, adolescents, and young adults represent a distinct patient group with similar characteristics that can benefit from immune checkpoint inhibitor therapy [PMID: 39701117]. Based on the above analysis, only two patients may have uncertain diagnoses with an extremely low probability of CMMRD, so they are not ruled out.

Additionally, we reviewed the mutation detection results in patients with ultra-high TMB (TMB > 300 muts/Mb). All of them included *POLE* gene somatic mutations. We have made detailed revisions regarding the limitations in our study.

In the future, it would be valuable to collect more confirmed CNS CMMRD and LS patients to explore and differentiate their characteristics.

We have revised the limitations in the **Discussion** section:

Second, constitutional mismatch repair deficiency (CMMRD) has been identified in individuals with biallelic pathogenic germline mutations in MMR genes [57,58]. In our study, among the 20 patients under the age of 25, two patients had two germline variants in the same MMR gene (*MLHI* LP and VUS; *PMS2* P and LP, respectively), which could not exclude the diagnosis of CMMRD. It has been reported that primary mismatch repair-deficient gliomas in children, adolescents, and young adults represent a distinct patient group with similar characteristics that can benefit from immune checkpoint inhibitor therapy [PMID: 39701117]. Given the small number of only two cases with an extremely low probability of CMMRD, we have not excluded them at this stage. In the future, the newly published method [PMID: 36240479; PMID: 38531023] can be considered to collect more confirmed cases of CNS CMMRD and LS according to the current differential diagnosis recommendations of CMMRD [PMID: 39420201], and explore and distinguish their characteristics, which have important clinical value. However, our study was retrospective and did not include family verification, so the diagnosis of CMMRD of the two patients could not be conclusively confirmed.

[Copy edits: Line 234 = should be "carriers" not carries]

Response: Thank you very much for your reminder. We have already corrected the misspelled word to the proper term “carriers” .

Reviewer #2 (Remarks to the Author):

This study describes a cohort of tumours diagnosed in a cohort of Lynch syndrome patents from China. Although the analysis of 238 tumours is relevant for the field, the manuscript can be certainly improved.

Please find below a list of comments/suggestions:

The overall contribution of the study is unclear, as well as its relevance to screening management within Lynch syndrome.

Response: Thank you very much for your kind reminder. In the previous version, we may not have effectively highlighted the focus of our study. We have now made revisions as suggested.

As mentioned in the **Introduction** section, tumors in the CNS related to LS have been less studied, with no large-scale cohort studies or screening guidelines available. The same applies to non-LS-related tumors. Therefore, our study divided the cases into three groups based on the LS tumor spectrum and tumor location. Our study aimed to investigate the similarities and differences in clinical and genomic features across multiple cancer types in CNS LS-related tumor and non-CNS LS-unrelated tumor groups compared with those in non-CNS LS-related tumor groups, which will improve LS screening, diagnostics, surveillance, prevention, management and therapy.

Regarding LS screening recommendations, our study may only provide some preliminary conclusions. CNS tumors, as an extracolonic manifestation of LS, appear to be relatively infrequent, with a quadruple occurrence risk. Our research shows that, compared to non-CNS LS-related tumors, CNS LS-related tumors are relatively rare, may manifest at an earlier age, but have similar characteristics of MMR deficiency and exhibit slight genetic heterogeneity. Similarities include a higher mutation burden. There was no significant difference in the TMB distribution between the CNS LS-related tumor group and the non-CNS LS-related tumor group. However, differences exist due to the tumor location. The cumulative lifetime risk of brain tumors is significantly greater in individuals with *MSH2* mutations than in those with variants in other MMR genes. Both the mean age (38 years) and the age of patients harboring *MSH6* and *PMS2* germline P/LP variants in the CNS LS-related tumor group were significantly lower than those in the other two groups. The CNS LS-related tumor group had a lower proportion of MSI-H tumors than the non-CNS LS-related tumor group. Our research serves as a fundamental reference for the diagnosis and treatment of CNS LS-related tumors.

In the non-CNS LS-unrelated tumor group in our study, the proportion of LS patients among all cancer types in this group was low, with a maximum of 0.6%. The mean age was 55.5 years. We also suggest relevant screening, with a potentially later screening age. The proportion of patients with *PMS2* (43.4%) was the highest. Our findings suggest that all LS patients should be concerned about the risk of LS-unrelated tumors, especially those carriers *PMS2* gene. This group had the fewest mutations, and the TMB in this group was significantly lower than that in the other two groups. The proportion of MSI-H was also lower.

Our findings suggest that non-CNS LS-unrelated tumors should not be grouped together with non-CNS LS-related tumors for genetic counseling or clinical management, and the later onset of non-CNS LS-unrelated tumors may also justify specific guidelines for surveillance that are tailored to this tumor type. Revised international clinical guidelines for MMR carriers for different tumor groups should be based on a comprehensive assessment of their associated cancer risks.

The clarity of the writing can be certainly improved. The terminology is very confusing in multiple instances: incidence/prevalence, mutation frequency of the allelic gene, FASTQ document, healthy allele, super high TMB....

Response: Thank you for the reminder. You are right that some parts were not precise enough. We have made the necessary revisions.

Abstract

The abstract lacks specificity in the presentation of numerical data, detailed methodology, and precise definitions, which limits the clarity and impact of the findings.

Response: Thank you very much for your valuable comments and suggestions. We have optimized the **Abstract** and added more details, which are outlined as follows:

Background: The characteristics may differ among LS patients with different tumor types. CNS LS and LS-unrelated tumors represent the uncharacterized tumor type in LS, and there is no sufficient consensus on guidelines and management recommendations. Therefore, additional studies are needed to elucidate the features of LS patients with noncanonical tumors.

Methods: In this study, we conducted a retrospective analysis of a substantial Chinese LS cohort to elucidate the clinical and molecular characteristics across various tumor types. A total of 238 pan-cancer samples from patients with LS were finalized enrolled and categorized into three groups according to the tumor location and LS tumor spectrum: central nervous system (CNS) LS-related tumor group (n=68), Non-CNS LS-related tumor group (n=117), and Non-CNS LS-unrelated tumor group (n=53).

Results: Our findings revealed significant disparities among the LS incidence in patients with endometrial cancer (p=0.0047) and liver cancer (p=0.011) when comparing our cohort with The Cancer Genome Atlas (TCGA) cohort. Moreover, tumor samples exhibited heterogeneity in the distribution of germline pathogenic genes and age at diagnosis across the three groups. The CNS LS-related tumor group had a lower age at diagnosis compared to the other two groups (p < 0.001). The proportions of *MSH2* in the CNS LS-related tumor group, *MLH1* in the Non-CNS LS-related tumor group and *PMS2* in the Non-CNS LS-unrelated tumor group were the highest. Furthermore, notable differences were observed in somatic mutation profiles, TMB, and MSI status. The TMB features in the CNS and non-CNS LS-related tumor groups were more similar and were significantly lower in the non-CNS LS-unrelated tumor group (p < 0.001). The Non-CNS LS-related tumor group had a greater proportion of MSI-H tumors than did the other two groups (p < 0.001).

Conclusions: Our study is pioneering in categorizing a large cohort of LS patients with pan-cancer into three groups based on tumor location and LS tumor spectrum, which further confirmed that LS was a highly heterogeneous tumor syndrome. The comprehensive description of the clinical manifestations and tumor characteristics in different disease sites and pathogenic genes enriched the reference data for LS, thereby providing evidence for future disease management strategies.

How is the prevalence of Lynch syndrome estimated from TCGA data?

Response: Thank you for bringing this to our attention. You are absolutely right—we did miss this point. We have now added the method used by TCGA to diagnose Lynch syndrome in the Methods section. The details are as follows:

Lynch syndrome analysis in The Cancer Genome Atlas (TCGA) cohort

We obtained data from the publication by Huang et al. [PMID: 29625052], including information on cancer types, germline genes with P/LP variants, specific mutations and variant classifications and so on. From this dataset, we identified samples with germline variants in MMR genes, which are associated with LS. We then calculated the incidence of LS across different cancer types and further compared these incidences with those in our own cohort.

Introduction

The introduction is too extensive and needs to be shortened. It does not help to understand the focus of the study and the specific objectives.

The topics in the introduction are not clearly defined, with concepts mixed in a way that makes it difficult to follow.

Response: Thank you very much for your valuable suggestions. We have condensed the Introduction and highlighted the main themes and key points, which will facilitate a clearer understanding.

Please refer to relevant and updated bibliography (i.e. ref 32 and 33 cannot be used as current LS guidelines)

Response: Thank you for pointing this out. Regarding references 32 and 33, we intend to highlight that there are no definitive guidelines for LS in the CNS. It seems our original wording was not clear enough. We have now revised it as follows:

Although LS quadruples the risk of brain tumors occurring, brain tumors are relatively rare and represent the uncharacterized tumor type in LS, and there is no sufficient consensus on guidelines and management recommendations [32,33].

One relevant topic is the specific characteristics of MSI in CNS LS-associated tumors (PMID: 36240479). Please discuss the findings according to this previous knowledge throughout the manuscript.

Response: Thanks for your suggestion. We have added this topic to the **Discussion** section, as follows:

In the CNS LS-related tumor group, MSI-H tumors were found only in samples with MLH1 (6/10) and MSH2 (11/29) germline P/LP variants. The CNS LS-related tumor group had a lower proportion of MSI-H tumors than the Non-CNS LS-related tumor group. This finding is consistent with previous reports indicating that MMRDness scores vary across different tissues, with brain tissue typically exhibiting lower scores compared to blood and gastrointestinal tissues [PMID: 36240479].

Methodology

In general, the terminology is not accurate.

Patients and samples section lacks clarity. Please clarify the type of samples analyzed, how LS patients are identified, type of mutations (germline/somatic?), sample inclusion / exclusion, etc...

Response: Thank you very much for your valuable suggestions. We have carefully revised the corresponding sections in the Methods, particularly the part on Patient Enrollment. The detailed revisions are as follows:

A total of 229 patients diagnosed with LS harboring heterozygous germline P/LP variants in at least one of the MMR genes (MLH1, MSH2, MSH6, and PMS2) were enrolled in this study. The 239 tumor samples from these 229 LS patients included 19 samples from 9 patients with multiple primary cancers. One sample from a patient with cancer of unknown primary origin was excluded. Considering that samples from different sites in patients with multiple primaries might be assigned to different groups, 238 samples from 228 LS patients were ultimately enrolled and used for all subsequent analyses. Baseline characteristics information, including age, gender and cancer types of all patients were collected. Among the 228 enrolled patients, 132 were male and 96 were female, with a median age of 52.5 years (range: 0–79).

What is Simceredx database?

Response: The Simceredx database is a germline genetic alteration frequency database for tumors, established by Simcere Diagnostics based on the Chinese population.

Please detail the genes included in the NGS analysis.

Response: Thanks for your reminder. We have added the details to the **Methods** section. The specific gene list can be found in the **Supplementary materials**.

The bioinformatics analyses are hard to follow. It is not clear how they have used the germline and somatic data, and which strategy has been used for variant calling (tumor-normal paired, tumor-only ...), which genes/regions have been analyzed...

Response: Thank you for your insightful comments. We have thoroughly revised the bioinformatics analyses section in the Methods. Specifically, we employed a tumor-normal paired strategy for variant calling. The specific gene list can be found in the **Supplementary materials**.

The terminology is not accurate. Lack of details on some tools that have been used (eg. MSI), filters and annotation.

Response: Thank you for the reminder. We have now added a relevant reference on MSI [PMID: 24371154] and removed the imprecise filters and annotations.

Can TMB be calculated from panel data? Has it been previously validated?

Response: Yes, we calculated TMB using a targeted gene panel. We have compared its consistency with the gold standard, whole-exome sequencing (WES), and the concordance rate was 93.87%.

Please specify the thresholds used for TMB and MSI categorization.

Response: Thank you very much for your prompt. Detailed explanations of the thresholds have been added to the gene list in the **Supplementary materials**.

The TCGA cohort is not described.

Response: Thank you very much for your reminder. We have now added a detailed description of the TCGA cohort in the Methods section. The specific details are as follows:

Lynch syndrome analysis in The Cancer Genome Atlas (TCGA) cohort

We obtained data from the publication by Huang et al. [PMID: 29625052], including information on cancer types, germline genes with P/LP variants, specific mutations and variant classifications and so on. From this dataset, we identified samples with germline variants in MMR genes, which are associated with LS. We then calculated the incidence of LS across different cancer types and

further compared these incidences with those in our own cohort.

It is not clear how they identify germline PV for identifying LS individuals (both in Sinceredex and TCGA cohorts)

Response: Thank you for your reminder. In the Sinceredex cohort, we determined germline P/LP variants according to the ACMG guidelines. We have added the specific details in the Methods section as follows:

According to the American College of Medical Genetics and Genomics (ACMG) guidelines, germline mutations were stratified into 5 categories, P, LP, benign, likely benign, and variants of unknown significance (VUS), in which P and LP variants are typically selected for further analysis [PMID: 25741868].

In the TCGA cohort, we directly referenced the data on germline P/LP variants from the cited article. Upon reviewing the specific descriptions in the article, we found that their classifications were also based on the ACMG guidelines [PMID: 29625052].

Results

The results section and figures are redundant.

The titles of the results are not self-explanatory and do not provide clear information.

Section 1 (“Patient enrollment”)

The reasoning behind the tumor classification into the three groups is not explained.

Response: Thank you for your valuable reminder. A total of 238 samples from 228 pan-cancer patients with LS were enrolled and divided into 3 groups according to the tumor location and tumor spectrum of the LS: the CNS LS-related tumor group, the Non-CNS LS-related tumor group, and the Non-CNS LS-unrelated tumor group. We have added clearer content in the **Introduction** to explain the rationale for the grouping, as detailed below:

LS increases the risk of developing several cancers throughout life, including cancers of the colon, rectum, endometrium, stomach, small bowel, biliary tract, pancreas, renal pelvis and/or ureter, bladder, kidney, ovary, brain, or prostate, which are known as LS-associated cancers because of their significantly greater frequency compared to that of the average population.

Although LS quadruples the risk of brain tumors occurring, brain tumors are relatively rare and represent the uncharacterized tumor type in LS, and there is no sufficient consensus on guidelines and management recommendations.

In addition, patients with LS can also develop LS-unrelated cancers, such as breast cancer and sarcomas, because the data are insufficient to demonstrate that the risk of developing these cancers is increased in individuals with LS.

Age, MSI level, genomic alterations in tumors and other characteristics may differ among LS patients with different tumor types; thus, additional studies are needed to address the features of LS patients with noncanonical tumors. Although some non-CNS LS-unrelated tumors and CNS LS-related tumors have been reported to develop in a few patients with LS and have been published as case reports and small retrospective cohorts, systematic studies with large sample sizes are lacking to describe the comprehensive characteristics of the population, especially comparisons with non-CNS LS-related tumors.

Why is the frequency of CNS-related LS tumors so high in the cohort?

Response: Thank you very much for your question. We reviewed the incidence proportion of LS in CNS tumors within our cohort, which was 1.2% (68 out of 5,655). In comparison, the overall

incidence proportion in the TCGA cohort was 0.55% (5 out of 908). **Figure 1E** illustrates the comparison of LS incidence rates between our cohort and the TCGA cohort. There was no statistically significant difference in the incidence rates of CNS LS between the two cohorts. The higher number of CNS patients with LS may be related to the larger baseline number of CNS tumors in our cohort.

The high proportion of CNS tumor patients under 20 suggests CMMRD. CMMRD can be discarded by measuring MSI in normal tissues by high sensitivity MSI analysis (i.e. PMID: 38531023)

Response: Thank you very much for your reminder. We have reviewed the germline variants (including Pathogenic, Likely Pathogenic, and Variant of Uncertain Significance) of the mismatch repair (MMR) genes detected in all patients under the age of 25 (N = 20). Except for two patients who had two germline variants in the same MMR gene (*MLH1* LP/VUS and *PMS2* P/LP, respectively), the remaining 18 patients each had only a single P/LP mutation in a definite gene.

In our retrospective study, there was indeed no way to obtain additional samples to determine whether these two patients had LS or CMMRD. However, in gliomas, primary mismatch repair-deficient gliomas in children, adolescents, and young adults represent a distinct patient group with similar characteristics that can benefit from immune checkpoint inhibitor therapy [PMID: 39701117]. Considering this, only two patients may have uncertain diagnoses, and they weren't ruled out. In addition, these two patients represent a small proportion of the overall cohort.

Additionally, we reviewed the mutation detection results in patients with ultra-high TMB (TMB > 300 muts/Mb). All of them included *POLE* gene somatic mutations. We have made detailed revisions regarding the limitations in our study.

In the future, it would be valuable to collect more confirmed CNS CMMRD and LS patients to explore and differentiate their characteristics.

We have revised the limitations in the **Discussion** section:

Second, constitutional mismatch repair deficiency (CMMRD) has been identified in individuals with biallelic pathogenic germline mutations in MMR genes [57,58]. In our study, among the 20 patients under the age of 25, two patients had two germline variants in the same MMR gene (*MLH1* LP and VUS; *PMS2* P and LP, respectively), which could not exclude the diagnosis of CMMRD. It has been reported that primary mismatch repair-deficient gliomas in children, adolescents, and young adults represent a distinct patient group with similar characteristics that can benefit from immune checkpoint inhibitor therapy [PMID: 39701117]. Given the small number of only two cases with an extremely low probability of CMMRD, we have not excluded them at this stage. In the future, the newly published method [PMID: 36240479; PMID: 38531023] can be considered to collect more confirmed cases of CNS CMMRD and LS according to the current differential diagnosis recommendations of CMMRD [PMID: 39420201], and explore and distinguish their characteristics, which have important clinical value. However, our study was retrospective and did not include family verification, so the diagnosis of CMMRD of the two patients could not be conclusively confirmed.

It is unclear what is being compared to the TCGA, or how the LS patients in the cohort are being used in the analysis. TCGA data is not introduced.

Response: Thank you for your reminder. We compared the incidence rate of LS within each specific cancer type across the two cohorts. We have now added a detailed description of the TCGA cohort in the Methods section. The specific details are as follows:

Lynch syndrome analysis in The Cancer Genome Atlas (TCGA) cohort

We obtained data from the publication by Huang et al. [PMID: 29625052], including information on cancer types, germline genes with P/LP variants, specific mutations and variant classifications and so on. From this dataset, we identified samples with germline variants in MMR genes, which are associated with LS. We then calculated the incidence of LS across different cancer types and further compared these incidences with those in our own cohort.

Please revise axis title in Figure 1E.

Response: Thank you very much for your reminder. We have revised the axis title in **Figure 1E** as follows:

Table 1 refers to tumors, not patients as stated in the title. Please clarify.

Response: Thank you very much for your reminder. We have revised the title in **Table 1** as follows:

Table 1 Characteristics for the tumors of enrolled patients with Lynch syndrome in the three groups.

Section 2 (“Germline mutational feature analyses”)

Very vague description of the germline mutational spectrum of the gene. Why silent variants have been considered?

Response: Thank you very much for your reminder. In our study, we determined germline P/LP variants according to the ACMG guidelines. We have added the specific details in the Methods section as follows:

According to the American College of Medical Genetics and Genomics (ACMG) guidelines, germline mutations were stratified into 5 categories, P, LP, benign, likely benign, and variants of unknown significance (VUS), in which P and LP variants are typically selected for further analysis [PMID: 25741868].

In the germline mutational spectrum, silent variants were not taken into consideration. However, silent (synonymous) variants were included in the comparison of mutation types within the somatic mutation results.

Please clarify the meaning of “novel” and “non-novel” mutations.

Response: Thank you very much for your reminder. We have added an explanation for the terms “novel” and “non-novel” mutations. Specifically, “novel” indicates that these mutations were neither reported in the literature nor included in the database. The detailed description is as follows:

The number of germline P/LP variant genes in the 238 LS samples is shown in Figure 2A, including *MLH1* (n=52), *MSH2* (n=69), *MSH6* (n=60), and *PMS2* (n=57), which were classified as novel (n=47, 19.75%) or non-novel (n=191, 80.25%) mutations according to whether they were reported

or included in the database. Novel mutations indicate that these mutations were neither reported in the literature nor included in the database.

This section needs to be summarized.

Response: Thank you for your valuable suggestions. We have added a summary to this section, as detailed below:

In brief, we found that tumor samples in the three groups exhibited heterogeneity in the distribution of germline pathogenic genes (Figure 2A and 2B), and there were significant differences in age among the three groups, with the age of the CNS LS-related tumor group being lower than that of the other two groups, especially in patients harboring *MSH6* and *PMS2* germline P/LP variants (Table 1, Figure 2C).

Section 3 (“Somatic mutational landscape and feature analyses”)

Please clarify what the contribution of this analysis is.

Mutational data lacks clear interpretation of its clinical or biological significance. The comparison between groups is not well contextualized, and the writing could be more concise.

Response: Thank you for your insightful question. The analysis of somatic mutation genes and characteristics across the three groups allows us to examine the similarities and differences at the genomic level, thereby gaining a deeper understanding of their potential associations with tumor development and progression. Meanwhile, the analysis of TMB (tumor mutation burden) and MSI (microsatellite instability) helps us determine whether there are any significant differences in the common features of Lynch syndrome (LS) among the three groups of LS patients. We have revised the details in the main text.

Please clarify how the 50 top genes were selected (the most mutated ones)? How have the significant differences between groups been assessed?

Response: Thank you very much for your question. We ranked the mutations based on population mutation frequency and selected the top 50 mutation genes with the highest frequency to generate heatmaps. We did not assess significant differences in the top 50 mutation frequency genes across groups. Instead, we compared different mutation types and the differences in TMB among the groups. We have revised the details in the main text.

Please detail the specific mutations identified in the 50 top mutated genes. Are there recurrent variants?

Response: Thank you for your question. However, in our study, it was not possible to identify recurrent variants.

Terminology of mutation type is not accurate, and no conclusion has been drawn from this data.

Response: Thank you for your input. We have removed the results from that section.

The non-CNS LS-unrelated tumors have low TMB. They are probably sporadic tumors. Is the identification of LS in this tumor samples well-assessed?

Response: Thank you for the prompt. Throughout our study, the criteria for diagnosing LS remained consistent. Specifically, we defined LS as harboring heterozygous germline pathogenic or likely pathogenic (P/LP) variants in at least one of the mismatch repair (MMR) genes (*MLH1*, *MSH2*, *MSH6*, and *PMS2*). There were no differences in the identification of LS based on group allocation. Therefore, regardless of which group the patients were assigned to, those who met the criteria were confirmed as having LS, rather than being sporadic cases. While our study identified Lynch syndrome through molecular diagnosis in patients with non-CNS LS-unrelated tumors, whether to

expand the Lynch tumor spectrum to include these tumors still requires further mechanistic research for validation. Our study analyzed Lynch syndrome carriers in a large population, focusing on the genetic landscape and clinical features of atypical Lynch tumors, thereby providing a foundation for future research.

Regarding the 5 glioma samples with TMB>300 mut/Mb, have they somatic mutations in POLE/POLD1 genes?

Response: Thank you very much for your reminder. We have reviewed the somatic mutation detection results for the five patients, and indeed, each patient was found to harbor mutations in hypermutated *POLE/POLD1* genes. We have added an explanation in the manuscript as follows:

In the non-CNS LS-unrelated tumor group, the TMB was generally low. In the CNS LS-related tumor group, 5 glioma samples had a TMB greater than 300 but were MSS, and we found each patient harboring mutations in hypermutated *POLE/POLD1* genes.

Figure 4: it is not possible to see all the columns represented in the graph (i.e. PMS2 in the CMS LS-related tumors group, MSS)

Response: Thank you very much for your careful review and the valuable suggestions you provided. We have revised **Figure 4A** to more clearly display all the columns represented in the graph, as shown below:

Section 4 (“The exceptional cases in LS”)

Hard to follow. Which germline mutations co-occur in the LS patients of the cohort?

Response: Thank you very much for your prompt. We have added descriptions of the specific genes in which germline mutations co-occur in the main text and provided detailed mutation information in the **Supplementary** file, as follows:

These co-occurring mutated genes were primarily involved in the DNA damage repair pathway, including *ATM*, *ATR*, *AXIN2*, *ERCC2*, *FANCA*, *FANCD2*, *MUTYH*, *NF1* (**Figure 4D**).

Co-occurring germline susceptibility mutations are indeed very rare, but they do occur as previously reported [PMID: 39493880; PMID: 39061189; PMID: 36760809; PMID: 36003761].

Table 2: Does the table refer to multiple primary tumors? It is surprising that patient PLS076 harbors

3 lung tumors.

Response: Thank you for your question. Yes, **Table 2** refers to multiple primary tumors, which are more common in patients with LS than sporadic tumor. Patient PLS076 indeed had three lung tumors. This phenomenon is indeed very rare, but it does exist and has been documented in the literature [PMID: 37846760; PMID: 26871768].

Discussion.

It needs to be extensively revised according to the previous comments.

Why MSI is present in non-LS associated tumors from *MLH1* carriers?

Please clarify how obtained results impact on LS management strategies, as stated in the abstract.

Response: Thank you very much for your careful review and the valuable suggestions you provided. We have made extensively revised according to the previous comments.

In our study, we did observe the phenomenon of MSI-H in non-LS-related tumors among *MLH1* carriers. We also reviewed the relevant literature. According to the reports [PMID: 30376427], in pan-cancer studies, MSI-H is indeed more frequently associated with *MLH1* carriers. The detailed results are as follows:

As mentioned in the **Introduction** section, tumors in the CNS related to LS have been less studied, with no large-scale cohort studies or screening guidelines available. The same applies to non-LS-related tumors. Therefore, our study divided the cases into three groups based on the LS tumor spectrum and tumor location. Our study aimed to investigate the similarities and differences in clinical and genomic features across multiple cancer types in CNS LS-related tumor and non-CNS LS-unrelated tumor groups compared with those in non-CNS LS-related tumor groups, which will improve LS screening, diagnostics, surveillance, prevention, management and therapy.

Regarding LS screening recommendations, our study may only provide some preliminary conclusions. CNS tumors, as an extracolonic manifestation of LS, appear to be relatively infrequent, with a quadruple occurrence risk. Our research shows that, compared to non-CNS LS-related tumors, CNS LS-related tumors are relatively rare, may manifest at an earlier age, but have similar characteristics of MMR deficiency and exhibit slight genetic heterogeneity. Similarities include a higher mutation burden. There was no significant difference in the TMB distribution between the CNS LS-related tumor group and the non-CNS LS-related tumor group. However, differences exist due to the tumor location. The cumulative lifetime risk of brain tumors is significantly greater in individuals with *MSH2* mutations than in those with variants in other MMR genes. Both the mean age (38 years) and the age of patients harboring *MSH6* and *PMS2* germline P/LP variants in the CNS LS-related tumor group were significantly lower than those in the other two groups. The CNS LS-related tumor group had a lower proportion of MSI-H tumors than the non-CNS LS-related tumor group. Our research serves as a fundamental reference for the diagnosis and treatment of CNS

LS-related tumors.

In the non-CNS LS-unrelated tumor group in our study, the proportion of LS patients among all cancer types in this group was low, with a maximum of 0.6%. The mean age was 55.5 years. We also suggest relevant screening, with a potentially later screening age. The proportion of patients with *PMS2* (43.4%) was the highest. Our findings suggest that all LS patients should be concerned about the risk of LS-unrelated tumors, especially those carriers *PMS2* gene. This group had the fewest mutations, and the TMB in this group was significantly lower than that in the other two groups. The proportion of MSI-H was also lower.

Our findings suggest that non-CNS LS-unrelated tumors should not be grouped together with non-CNS LS-related tumors for genetic counseling or clinical management, and the later onset of non-CNS LS-unrelated tumors may also justify specific guidelines for surveillance that are tailored to this tumor type. Revised international clinical guidelines for MMR carriers for different tumor groups should be based on a comprehensive assessment of their associated cancer risks.

With Kind Regards,

Zhaobang Tan and Xing Zhang

We would like to resubmit our manuscript entitled "**The clinical and genomic features of Lynch syndrome in distinct tumor spectrum groups: Implications for clinical practice**" to your journal. I appreciate the time and effort invested by the reviewers in evaluating my work. Thanks to the reviewers for their valuable comments. We have looked over the comments carefully and have made corrections which we hope will be able to approve. The following is our response after careful consideration:

Response to reviewers' comments:

Reviewer #1 (Remarks to the Author): Lynch syndrome clinical genetics

Thank you very much for addressing reviewers' concerns with your revised manuscript. However, before publication, the following changes need to be made:

While I appreciate the changes made to reflect reviewers' comments in the abstract, the abstract background needs to be changed to read more cleanly, for example:

"Lynch Syndrome (LS) patients may sometimes develop cancers of the central nervous system (CNS) as well as in sites not traditionally associated with LS. These tumors are poorly characterized in the literature, and there are no sufficient consensus on guidelines and management recommendations for these tumors. Therefore, additional studies are needed to elucidate the features of LS patients with noncanonical tumors."

✚ **Response:** Thank you for your insightful suggestion. We have revised the background of the **Abstract** (Line 36-40) as you recommended, and the changes have indeed made the **Abstract** background clearer and the **Abstract** more coherent. We sincerely appreciate your guidance.

-Line 42 = what does finalized mean?

✚ **Response:** Thank you for the reminder. We have revised the details and removed the ambiguous wording (Line 43).

-Line 49 = should be "germline pathogenic variants"

✚ **Response:** Thank you very much for your reminder. We have carefully revised the details in accordance with your suggestions (Line 49).

-Line 420/421 = Remove “The incidence of primary tumors is relatively low in clinical practice” as this statement seems surplus to requirements; moreover, depending on one’s definition of cancer, 1 in 2 individuals ultimately develop some sort of cancer

✚ Response: Thank you very much for your suggestion. We have carefully revised the relevant sections of the **Discussion (Line 436)**, removing the inappropriately phrased sentences as advised.

Reviewer #3 (Remarks to the Author): Lynch syndrome computational

As a new reviewer on this manuscript, I was asked to primarily respond to the revisions made based on the comments raised by reviewer #2, but also to give general feedback on the study. Regarding the comments of reviewer #2, I acknowledge that the paper has been improved and many points raised have been properly addressed. There are, however, als some criticisms that remain unaddressed or incompletely addressed:

1. R#2: “The overall contribution of the study is unclear as well as its relevance to screening management within Lynch syndrome.” In my opinion, this aspect is only partly addressed, particularly regarding to screening guidelines. For defining strategies for screening it is crucial what the risk is for carriers of MMR mutations to develop a particular type of cancer. Such a study requires a perspective from the family or patient, not from the tumor. A characterization of tumors, as done in this study, provides information on relative differences of tumor types between the MMR genes and about the likelihood that the underlying germline mutations are indeed causative, for example by presenting with high TML or MSI. It is certainly true that the age at diagnosis of a particular tumor type may be helpful in developing criteria for screening, but whether a screening for a cancer type is needed requires data on the incidence of that cancer. Therefore, it is desirable that the authors are more precise on how their data contribute to improve LS surveillance.

✚ Response: We sincerely thank the reviewer for this thoughtful comment. We agree that tumor characterization alone cannot directly define screening strategies. We acknowledge that our study was not designed to evaluate cancer incidence, lifetime risk, or penetrance in MMR variant carriers, which are essential parameters in determining screening strategies for Lynch syndrome (LS). Our research was conducted from a tumor-centric

perspective, focusing on the characterization of clinical and genomic features across tumor spectrum groups rather than on patient- or family-based risk estimation.

- ✚ To avoid overinterpretation, we have removed or revised all statements related to potential implications for LS screening strategies from the **Introduction, Results** and **Discussion** sections. Instead, we have clarified that our findings may serve as a reference for future research aiming to refine surveillance approaches, but they do not directly inform screening recommendations.

We greatly appreciate the reviewer's guidance in helping us improve the clarity and scientific focus of our manuscript.

2. R#2: "Please refer to relevant and updated bibliography (i.e. ref 32 and 33 cannot be used as current LS guidelines)". This comment refers to refs 24 and 25 in the revision. I agree with the reviewer that these references are not optimal. Please check e.g. the ERN Genturis website for the latest published recommendations for LS.

- ✚ **Response:** Thank you very much for pointing this out. We have added and updated our references and now cite the most recent and authoritative guidelines, including the ERN GENTURIS website cited recommendations [References 19,23,27], the NCCN Clinical Practice Guidelines (Version 1.2025) [References 20,24], and the ESMO guideline on hereditary gastrointestinal cancers [Reference 17] (Line 95-97).

3. R#2: "The high proportion of CNS tumor patients under 20 suggests CMMRD. CMMRD can be discarded by measuring MSI in normal tissues by high sensitivity MSI analysis (i.e. PMID: 38531023)". This point is not properly addressed. There are two patients in their cohort with two germline mutations in a MMR gene. I do not agree with the authors that CMMRD is unlikely in these cases. In fact, with two (likely) pathogenic mutations in PMS2 it is very likely that this patient has CMMRD and also for the MLH1 biallelic carrier (P + VUS), it is highly suspected. These patients will certainly have high tumor mutational loads and, in case of a CNS tumor, will likely also have a somatic POLE or POLD1 on top of that (PMID 40368937). This should be visible in the data. The CMMRD tumor spectrum is different between CMMRD and LS, so every reason to take them out. The statement in the discussion that the two cases have "an extremely low probability of CMMRD" cannot be based on their data and is simply wrong.

- ✚ **Response:** We thank the reviewer for raising this critical and insightful point. We fully agree that the two cases (PLS081 with *PMS2* P + LP and PLS227 with *MLH1* LP + VUS)

are suspicious for CMMRD. In the revised manuscript we now explicitly state that these two patients had two germline variants in the same MMR gene (PLS081 *PMS2* P and LP; PLS227 *MLH1* LP and *VUS* respectively), which could not exclude the diagnosis of CMMRD.

Upon reflection, we acknowledge that our original wording was misleading. Our intention was not to suggest that these two patients themselves had a low likelihood of CMMRD, but rather that the overall incidence of CMMRD is very low in the general population, with approximately one in a million if parents are not related. These two patients represent a small proportion of the overall cohort. “an extremely low probability of CMMRD” The statement is indeed flawed. To eliminate any misunderstanding, we have replaced “probability” with “population incidence,” clarified the sentence, and removed the potentially misleading phrase.

We have further acknowledged the inherent limitations of our study, and we also added references to recent CMMRD diagnostic guidelines (e.g., Marin et al., 2024, Clin Chem; Colas et al., 2024, Eur J Hum Genet [References 60,61]) and emphasized the value of employing high-sensitivity constitutional MSI assays in future studies. Revised text in **Discussion** as follows (Line 445-456):

In our study, among the 20 patients under the age of 25, two patients had two germline variants in the same MMR gene (PLS081 *PMS2* P and LP; PLS227 *MLH1* LP and *VUS* respectively), which could not exclude the diagnosis of CMMRD. It has been reported that primary mismatch repair-deficient gliomas in children, adolescents, and young adults represent a distinct patient group with similar characteristics that can benefit from immune checkpoint inhibitor therapy⁵⁹. Given the small number of only two cases with an extremely low population incidence of CMMRD, we have not excluded them at this stage. In the future, the newly published method^{48,60} can be considered to collect more confirmed cases of CNS CMMRD and LS according to the current differential diagnosis recommendations of CMMRD⁶¹, and explore and distinguish their characteristics, which have important clinical value. However, our study was retrospective and did not include family verification, so the diagnosis of CMMRD of the two patients could not be conclusively confirmed.

✚ We further confirmed the details of the two cases:

- PLS227 – age 12, MSS, 30 somatic mutations, no *POLE/POLD1* alterations.
- PLS081 – age 12, TMB 10.38, MSS, 15 somatic mutations, no *POLE/POLD1* alterations.

✚ These patients do not overlap with the six glioma samples exhibiting ultra-high TMB (>300 mut/Mb, MSS) and *POLE/POLD1* hypermutation described in **Results** as follows (Line 299-302):

In the CNS LS-related tumor group, 6 glioma samples (PLS041, PLS046, PLS124, PLS143, PLS185, PLS228) had a TMB greater than 300 but were MSS, and we found

each patient harboring mutations in hypermutated *POLE/POLD1* genes.

- ✚ Finally, we have added **Source data tables** and explicit patient identifiers throughout the manuscript to improve transparency and reproducibility. We are grateful to the reviewer for helping us clarify this important issue.

4. I would not prefer the use of “non-novel” mutations. Previously identified/known would be better.

- ✚ **Response:** We thank the reviewer for this wording suggestion. We have revised the terminology throughout the manuscript (**Line 221**), **Figure 2** and **Source data**, replacing “non-novel” with “previously reported” or “known” to enhance clarity and precision.

5. R#2: “Section 3 (“Somatic mutational landscape and feature analyses”) Please clarify what the contribution of this analysis is. Mutational data lacks clear interpretation of its clinical or biological significance.” The value of comparing mutation spectra of different tumor types with each other is not made clear. How can a comparison of heatmaps based on mixed groups of tumor types provide ‘deeper understanding of their potential associations with tumor development and progression.’? What is this deeper understanding? It is known that the spectrum of CNS tumors is different from non-CNS Lynch spectrum tumors, but there are also differences between endometrial and colorectal cancers. The tumors in the non-related group is even more diverse.

- ✚ **Response:** We sincerely appreciate this important observation. Our study is inherently retrospective and descriptive; it was not designed to uncover mechanistic underpinnings or to establish direct clinical correlations.

- ✚ The primary contribution of this analysis is to systematically catalog and compare the somatic mutational landscape across Lynch syndrome tumors with distinct tissue origins and clinical spectra, using a relatively large cohort of pan-cancer LS cases. Specifically (**Line 333-337**):

In this study, 238 final enrolled samples from 228 patients were divided into 3 groups according to tumor location and tumor spectrum of LS: the CNS LS-related tumor group (n=68), the Non-CNS LS-related tumor group (n=117), and the Non-CNS LS-unrelated tumor group (n=53). Our study analyzed clinical and genomic characteristics and the similarities and differences among the 3 groups.

To our knowledge, such stratified data are scarce in the literature, particularly in Asian populations. By comparing clinical features and genomic parameters across these groups, we filled a notable data gap—especially for Asian populations—regarding LS heterogeneity.

- ✚ These descriptive data highlight differences in mutation burden, gene-level alterations, and MSI status between LS-related and LS-unrelated tumors, and among different germline MMR variants. We did not attempt to draw cancer-type-specific conclusions; instead, we focused on the shared genomic signatures of each group and the inter-group differences. The value of comparing the mutation spectra of different groups with each other is as follows (Line 419-425):

Significant differences in genomic mutation profiles, mutation types and genes with high frequencies of mutations among the three groups were also observed. This observation implies that LS exhibits significant heterogeneity, indicating that its diverse pathogenesis is driven by distinct genetic alterations. The observed differences in age at onset, MSI, and TMB can serve as pilot data for future risk-prediction models and guideline revisions. This study offers valuable reference points for future mechanistic investigations and clinical decision-making, providing a robust hypothesis-generating foundation for both basic research and prospective trials.

- ✚ We agree that these results are hypothesis-generating rather than conclusive. Accordingly, we have revised the corresponding sections to frame the mutational analysis as a foundation for future functional studies and prospective clinical investigations. We appreciate the reviewer’s guidance in helping us position our findings within the appropriate scientific context.

6. R#2: “Please detail the specific mutations identified in the 50 top mutated genes. Are there recurrent variants?” The authors state that it was not possible to identify recurrent mutations, but it is unclear why. Perhaps the question was misunderstood, but in general, the study lacks openness about core data on which they base their conclusions.

- ✚ **Response:** Thank you for highlighting this. We have added a new **Source data** Table (Figure 3A-C) listing the specific somatic mutations found in the top 50 genes across the three subgroups. This includes variant types, genomic location, and allele frequency. Highly repetitive mutation sites were found in Non-CNS LS-related group and CNS LS-related group, as follows:

Group	Gene	Transcript	Hgvs_c	Hgvs_p	Variant_classification	Frequency
Non-CNS LS-related group	ACVR2A	NM_001278579.1	c.1310del	p.K437Rfs*5	Frame_Shift_Del	61.5% (72/117)
Non-CNS LS-related group	TCF7L2	NM_030756.4	c.1385del	p.K462Sfs*23	Frame_Shift_Del	35.0% (41/117)
Non-CNS LS-related group	MSH3	NM_002439.4	c.1148del	p.K383Rfs*32	Frame_Shift_Del	28.2% (33/117)
Non-CNS LS-	RNF43	NM_017763.5	c.1976del	p.G659Vfs*41	Frame_Shift_Del	25.6%

related group						(30/117)
Non-CNS LS-related group	WRN	NM_000553.4	c.725-9del	/	Intron	23.1% (27/117)
Non-CNS LS-related group	TGFBR2	NM_001024847.2	c.458del	p.K153Sfs*35	Frame_Shift_Del	22.2% (26/117)
CNS LS-related group	TP53	NM_000546.5	c.817C>T	p.R273C	Missense_Mutation	23.5% (16/68)
CNS LS-related group	PIK3CA	NM_006218.3	c.263G>A	p.R88Q	Missense_Mutation	20.6% (14/68)

The **Source data** table enhances transparency of the dataset and provides readers with the ability to examine gene-level alterations.

7. R#2: “The non-CNS LS-unrelated tumors have low TMB. They are probably sporadic tumors. Is the identification of LS in this tumor samples well-assessed?”. The reviewer raises an important point here, but the question may have caused confusion. Whereas the diagnosis of LS is beyond doubt is a reasonable assumption that the tumors are sporadic tumors, i.e. not caused/driven by MMR deficiency. LS families have higher risk of cancer, but obviously they also develop sporadic MMR-proficient cancers.

✚ **Response:** We appreciate the reviewer’s important observation and believe this may reflect a common point of misunderstanding in studies involving hereditary cancer syndromes. All patients included in our cohort were confirmed to carry pathogenic or likely pathogenic (P/LP) germline variants in MMR genes, based on ACMG/AMP standards, and therefore meet the diagnostic criteria for Lynch syndrome (LS).

✚ However, we fully agree that not all tumors arising in LS patients are necessarily driven by MMR deficiency. As correctly pointed out, the low TMB and MSS status observed in many of the non-CNS LS-unrelated tumors suggest that these may be sporadic tumors that are not mechanistically linked to the patients’ underlying LS mutations.

✚ To clarify, we have added a statement in the **Discussion** to acknowledge that (Line 382-385):

LS patients may develop sporadic tumors not associated with MMR deficiency. These tumors may represent background cancer risk unrelated to LS. Our tumor-based analysis describes features of tumors arising in LS carriers but does not imply that every tumor is causally related to LS.

We thank the reviewer for allowing us to clarify this key conceptual distinction.

8. R#2: “Regarding the 5 glioma samples with TMB>300 mut/Mb, have they somatic

mutations in *POLE/POLD1* genes?” It is nice that these specific data are now provided in the manuscript, but this, again, illustrates the incomplete presentation of the data. It is impossible for readers to correlate individual findings, like type of germline findings, somatic mutations, TMB, MSI status etc. Raw data should be made available (through EGA submission) and detailed supplementary tables should be provided that allow readers to correlate independent findings to patient IDs. Several of the points raised by reviewer #2 originate from this underlying shortcoming.

✚ **Response:** We fully agree with the reviewer. To enhance transparency, we have added a new **Source data** Table for all the pictures, which includes patient ID, tumor type, germline MMR variant, somatic mutations, TMB, and MSI status and so on. During this process, we identified a total of six glioma samples exhibiting ultra-high TMB (>300 mut/Mb, MSS) and *POLE/POLD1* hypermutation described in **Results** as follows (Line 300-302):

In the CNS LS-related tumor group, 6 glioma samples (PLS041, PLS046, PLS124, PLS143, PLS185, PLS228) had a TMB greater than 300 but were MSS, and we found each patient harboring mutations in hypermutated *POLE/POLD1* genes.

✚ We have revised the **Availability of data and materials** statement and deposited all raw data in a public repository. The updated details are as follows (Line 500-502):

The genomic sequencing data generated in this study have been deposited in the China National Center for Bioinformation (<https://ngdc.cncb.ac.cn/omix>, OMIX ID: OMIX009189). The remaining data are available in the Manuscript, Supplemental Materials and Source data. Source data are provided with this paper.

We thank the reviewer for pointing out this key issue related to data presentation and reproducibility.

In addition to the above comments, I believe that the findings presented in this study and the approach used in it are interesting. The study is descriptive but provides some new insights. Nevertheless, I do still have some major concerns:

- Data availability. I do not consider it acceptable that data are available upon request. Sequence data should be made available in EGA as is common practice in the current open science community. In addition detailed Tables are needed that couple all presented findings to patient IDs, to allow correlations between individual findings, which includes TMB, MSI status, germline mutations, driver mutations, etc to patient samples.

✚ **Response:** We sincerely thank the reviewer for emphasizing the importance of data

transparency. We fully agree that sequencing data should be made publicly available in accordance with open science principles. To enhance transparency, we have added a new **Source data** Table for all the pictures, which includes patient ID, tumor type, germline MMR variant, somatic mutations, TMB, and MSI status and so on. The detailed Tables can couple all presented findings to patient IDs, to allow correlations between individual findings, which includes TMB, MSI status, germline mutations, driver mutations, etc to patient samples.

- ✚ We have revised the **Availability of data and materials** statement and deposited all raw data in a public repository. The updated details are as follows (Line 500-502):

The genomic sequencing data generated in this study have been deposited in the China National Center for Bioinformation (<https://ngdc.cncb.ac.cn/omix>, OMIX ID: OMIX009189). The remaining data are available in the Manuscript, Supplemental Materials and Source data. Source data are provided with this paper.

This table allows full cross-comparison and correlation of data across samples. We appreciate the reviewer's guidance in helping us improve the reproducibility and usability of our dataset.

- The authors state in their discussion that ‘Our findings suggest that all LS patients should be concerned about the risk of LS-unrelated tumors, especially those carriers PMS2 gene.’ I believe that this statement is illustrative of a misconception in this study. PMS2 variants are common in the normal population. What is the evidence that these tumors caused by LS? In LS, the distribution of MMR genes is biased towards MLH1 and MSH2 which are the genes with higher penetrance. In the LS-unrelated group, the prevalence probably more resembles the prevalence of pathogenic MMR mutations as seen in the normal population, in which PMS2 variants are most prevalent. This fact does not exclude that tumors in the LS-unrelated group are caused by MMR deficiency, but it is not unlikely that in the majority of cases the tumors lack a second hit and are MMR-proficient. This is also what is observed by the authors and, therefore, the conclusion should be that these tumors are truly LS-unrelated.

- ✚ **Response:** We sincerely thank the reviewer for this important clarification. We agree that the high frequency of *PMS2* variants in the general population may account for its overrepresentation in the LS-unrelated group, and that MMR-proficient tumors in this group should not be automatically attributed to LS. We recognize that our original statement was imprecise and may have led to confusion.

- ✚ In response, we have revised the **Discussion** section to more accurately reflect this interpretation. We have removed the statement suggesting that all LS patients need to be “concerned” about LS-unrelated tumors. Instead, we highlight the need for further

studies to clarify which tumors in LS carriers are causally linked to MMR deficiency and which are not, as the reviewer insightfully suggests. Specifically (Line 382-393 and 402-404):

LS patients may develop sporadic tumors not associated with MMR deficiency. These tumors may represent background cancer risk unrelated to LS. Our tumor-based analysis describes features of tumors arising in LS carriers, but does not imply that every tumor is causally related to LS. *PMS2* variants have lower penetrance and higher prevalence in the general population. In the LS-unrelated group, the prevalence probably more resembles the prevalence of pathogenic MMR mutations as seen in the normal population, in which *PMS2* variants are most prevalent. The predominance of *PMS2* in the LS-unrelated group likely reflects background distribution, and that most of these tumors might be indeed “LS-unrelated” in biology. This fact does not exclude that tumors in the LS-unrelated group are caused by MMR deficiency, but it is not unlikely that in the majority of cases the tumors lack a second hit and are MMR-proficient. Future studies are needed to clarify which tumors in Lynch syndrome carriers are causally linked to mismatch-repair deficiency and which are not.

Non-CNS LS-unrelated tumors exhibit molecular features (late onset, low TMB/MSS) consistent with sporadic cancers not driven by MMR deficiency, which indicated that these tumors may be truly unrelated with LS.

With Kind Regards,

Zhaobang Tan and Xing Zhang

REVIEWERS' COMMENTS

Reviewer #3 (Remarks to the Author):

Thank you for addressing these points and clarifying the inconsistencies. Most of these issues are now more clearly formulated. As I indicated already, studying LS from the tumor perspective is interesting and important and the cohort is certainly informative, but it is essential that the difference between this approach and that of studies using the family/patient perspective is made clear to readers, and that the hypotheses and aims are adjusted to that.

✚ **Response:** We sincerely thank the reviewer for the constructive feedback and the careful reading of our revised manuscript. We appreciate the recognition of the importance of our work and the helpful suggestions for further improving the clarity of our manuscript. In the **Abstract, Introduction, and Discussion**, we have now explicitly framed and underscored that this study approaches Lynch syndrome from the tumor perspective, and adjusted the hypotheses and aim to that.

✚ Below, we provide point-by-point responses to the remaining comments:

There are a few small remarks remaining:

- Lines 229-302: "... harboring mutations in hypermutated POLE/POLD1 genes." Please rephrase, because the POLE/POLD1 genes themselves are not hypermutated.

✚ **Response:** We thank the reviewer for pointing this out. We have rephrased and revised the sentence.

Specifically:

...we found each patient harboring mutations in the DNA polymerase genes, POLE or POLD1, which are associated with a hypermutated phenotype.

- Discussion (lines 409-411): "...that non-CNS LS-unrelated tumors should not be grouped

together with non-CNS LS-related tumors for genetic counseling or clinical management”. Here, it is again the wording that is not chosen well. When we talk about genetic counseling we talk about patients not tumors. The patients have LS and should be counseled in the proper way irrespective of a cancer diagnosis or diagnosed tumor type.

✚ **Response:** We fully agree with the reviewer's concern. We have revised the wording to emphasize that genetic counseling pertains to patients rather than tumors.

Specifically:

People with LS should receive lifelong cancer screening starting in adulthood, and should be counseled in the proper way irrespective of a cancer diagnosis or diagnosed tumor type. Surveillance of non-CNS LS-related tumors such as CRC, EC, and urothelial cancer has been recommended for individuals with LS at an early age 16. However, there is not enough evidence to support increased screening above average risk in Non-CNS LS-unrelated tumor screening recommendations. Our findings suggest that patients with non-CNS LS-unrelated tumors should not be grouped together with patients with non-CNS LS-related tumors when considering genetic counseling or clinical management, and the later onset of non-CNS LS-unrelated tumors may also justify specific guidelines for surveillance that are tailored to this tumor type. Revised international clinical guidelines for MMR carriers for different tumor groups should be based on a comprehensive assessment of their associated cancer risks.

- Discussion (lines 445-456): “...which could not exclude the diagnosis of CMMRD..” still is a suboptimal way to phrase this. The fact that these patients had two LP/P variants strongly suggests a diagnosis of CMMRD. The phrasing 'does not exclude' suggests to readers that the authors have found something that argues against a CMMRD diagnosis, but without complete evidence.

✚ **Response:** We appreciate the reviewer's clarification. We have rephrased the statement to reflect that the findings suggest CMMRD rather than implying uncertainty.

Specifically:

In our study, among the 20 patients under the age of 25, two patients had two germline variants in the same MMR gene (PLS081 PMS2 P and LP; PLS227 MLH1 LP and VUS

respectively), which strongly suggests a diagnosis of CMMRD.

The above is our response after careful consideration of the reviewer's comments, please contact us if you have any questions.

Yours sincerely,

Zhaobang Tan and Xing Zhang